# Evidence for Alfvén waves powering auroral arc via a static electric potential drop

S. Tian [1,2] ✉, Z. Yao [3,4] ✉, J. R. Wygant[2], R. L. Lysak [2], J. Bortnik [1], L. R. Lyons[1], J. Liang[5], R. Shi[6], C. P. Ferradas[7,8], Y. Shen [9] & G. D. Reeves [10,11]

Natural light displays known as the aurora provide a captivating glimpse into the electromagnetic dynamics in space plasmas. Aurorae are not exclusive to Earth but also observed on celestial bodies including planets and even comets. Previous studies have unveiled two fundamental auroral acceleration mechanisms: electric potential and Alfvénic acceleration. However, the relation of the energy processes associated with the auroral acceleration region has remained unclear mainly due to the lack of quantitative analysis. Employing quantitative assessment of energy budget using multi-platforms from the magnetosphere to auroral ionosphere, our findings underscore the interplay between these two mechanisms. Here we show that energy carried by Alfvén waves travels from the magnetosphere to the auroral acceleration region, forming an electric potential drop that accelerates particles to produce aurorae. Similarities in auroral particle behaviors between Earth and Jupiter suggest the applicability of the terrestrial scenario to Jupiter and potentially other celestial bodies in the Universe.

Particle energization in the auroral acceleration region (AAR) is one of the major mysteries in planetary space environments[1–4]. Although the characteristics of accelerated particles were reported for decades[5–7], key aspects of the energization mechanism remain poorly understood. Based on the particle features, two acceleration pictures have been observed: monoenergetic acceleration (or inverted-V structure) produced by a quasi-static electric potential drop along magnetic field lines[5,8] and broadband acceleration produced by Alfvén waves[9–11]. The energy budget in the magnetosphere-ionosphere coupling and thus the partition to auroral particles is still unclear. The major challenges in understanding the particle acceleration mystery are the energy source and the energy conversion for these two mechanisms.

It is widely accepted that an upward-pointing electric field gives the intense downward monoenergetic accelerated electrons that are the main energy carriers that directly power bright auroral emissions.

However, the formation of this electric potential drop is a key mystery. Moreover, this potential drop acceleration was also identified at Jupiter by the Juno spacecraft, which arrived at Jupiter in 2016 to explore its auroral acceleration region in situ[12,13]. Figure 1 shows two typical examples of the potential drop acceleration of electrons at the Earth (left) and Jupiter (right). The inverted-V structures marked by the red arrows are typical features of potential drop acceleration[13]. Although the structures, showing the inverted-V shape, are similar at both planets, their energies are significantly different. Typical energies for potential drop acceleration are about several keV at Earth[5,6] and about 100 keV or higher at Jupiter[14]. The electric potential drop above these planetary polar auroral regions is analogous to a space capacitor, but the key question is: what controls the operation of the capacitor?

A key insight to solve the auroral acceleration problem is to understand the energy budget from the source region to the auroral

[1]Department of Atmospheric and Oceanic Sciences, University of California, Los Angeles, California, USA. [2]School of Physics and Astronomy, University of Minnesota, Minneapolis, MN, USA. [3]NWU-HKU Joint Centre of Earth and Planetary Sciences, Department of Earth Sciences, University of Hong Kong, Hong Kong SAR, China. [4]Department of Physics and Astronomy, University College London, London, UK. [5]Department of Physics and Astronomy, University of Calgary, Calgary, Canada. [6]School of Ocean and Earth Science, Tongji University, Shanghai, China. [7]Geospace Physics Laboratory, NASA Goddard Space Flight Center, Greenbelt, MD, USA. [8]Department of Physics, Catholic University of America, Washington, DC, USA. [9]Department of Earth, Planetary, and Space Sciences, University of California, Los Angeles, California, USA. [10]Los Alamos National Laboratory, Los Alamos, NM, USA. [11]The New Mexico Consortium, Los Alamos, NM, USA. ✉e-mail: ts0110@atmos.ucla.edu; yaozh@hku.hk

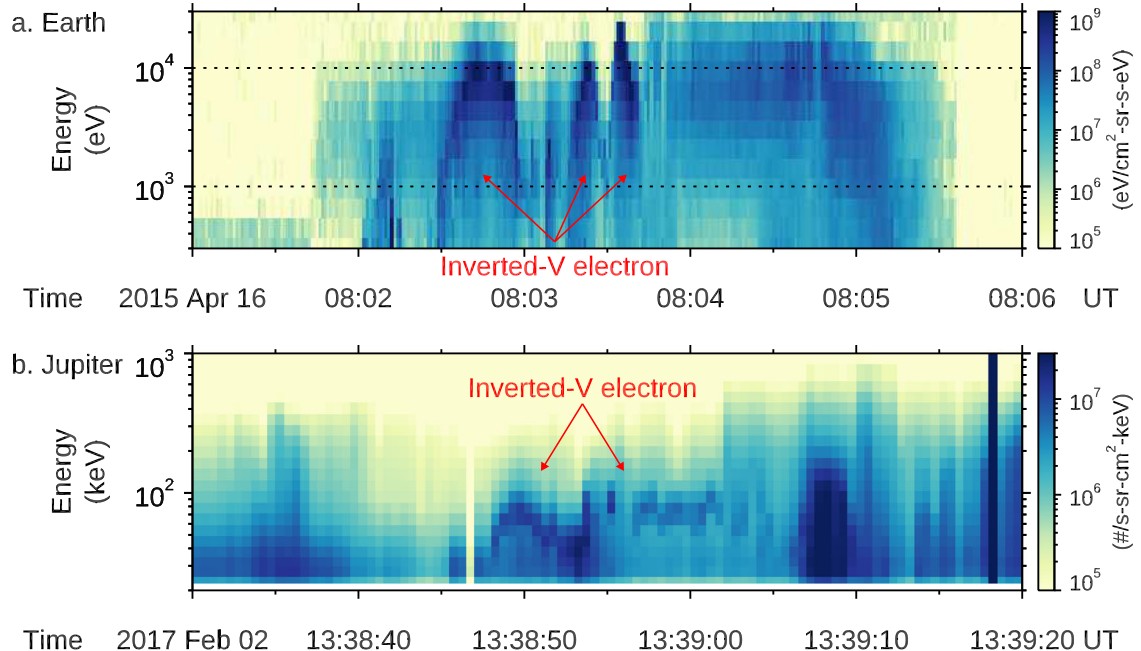

**Fig. 1 | The "inverted-V" electron signature produced by potential drop acceleration.** Electron spectra produced by potential drop accelerations at **a** Earth and **b** Jupiter. The terrestrial observation was from the SSJ instrument onboard the Defense Meteorological Satellite Program (DMSP) F19 spacecraft[76]. The Jovian observation was from the JEDI instrument onboard the Juno spacecraft, reproduced after Mauk et al.[13].

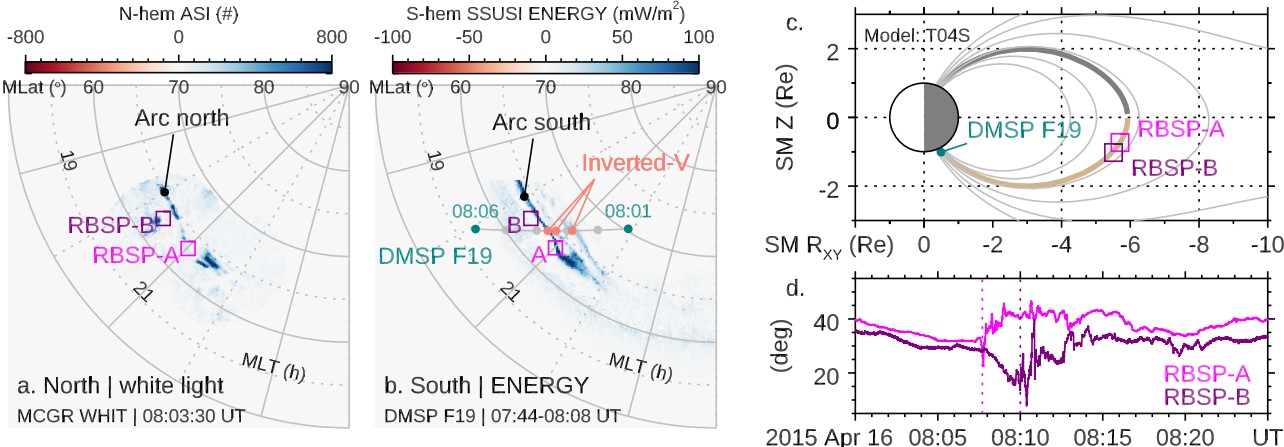

**Fig. 2 | The auroral images and the corresponding magnetosphere measurements. a** The auroral image taken by THEMIS ASI (white light, raw count) in the northern hemisphere. The image is presented in the plane of magnetic latitude (MLat) in degree (○) and magnetic local time (MLT) in hour (h). **b** The auroral image taken by DMSP F19 SSUSI (UV emission, calibrated to energy flux) in the southern hemisphere. The footprints of RBSP-A and B are mapped using the T04 storm-time magnetic model[77]. **c** The magnetic field lines connecting the magnetosphere and ionosphere. The locations of RBSP-A and -B and DMSP F19 are marked along the highlighted magnetic field line. **d** The elevation angle ($\theta = \sin^{-1} B_z / |B|$) of the magnetic field measured by the two RBSP satellites, showing strong perturbations when crossing the magnetic field lines threading the auroral arc.

acceleration region and to the auroral emission altitudes. So far, such comprehensive observations may only be achievable in the terrestrial system. One Earth auroral event on April 16, 2015 shown in Fig. 1 was captured by ground optical observations, low-altitude spacecraft for measuring electron precipitation above the aurora and the Van Allen Probes (also known as the Radiation Belt Storm Probes, RBSP) for tracking the magnetospheric energy source. This event occurred during an extended moderate geomagnetic storm. Details of the data and instruments are presented in **Methods, subsection Satellites and instrumentation.**

In this paper, the coordinated measurements provide a fortuitous opportunity to quantitatively evaluate the key energy conversion processes that lead to the formation of electric potential drop in the auroral acceleration region. We quantitatively evaluate the energy budget above and below the AAR and suggest that Alfvén waves pump electromagnetic energy to sustain the electric potential drop of the AAR.

## Results

### The energy flow above AAR

Figure 2a and b present coincident auroral images from the two hemispheres. The RBSP-A and -B satellites, located at [-4.0, 4.1, -0.7] $R_E$ (Earth Radii) and [-3.2, 4.6, −1.0] $R_E$ in Solar Magnetospheric (SM) coordinates (Fig. 2c), crossed the magnetic flux tubes connecting the auroral arc as illustrated by the footprints shown in Figs. 2a, b, and the meridional view of the magnetic field in Fig. 2c. The local magnetic

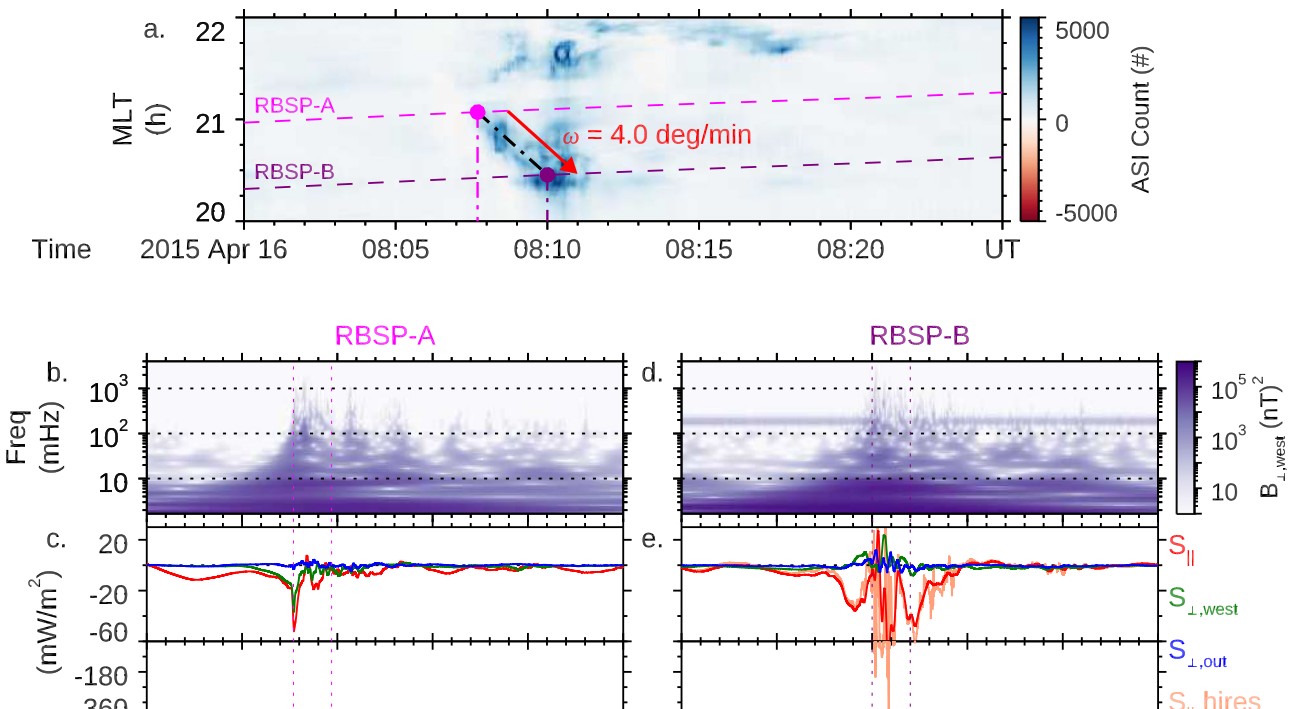

**Fig. 3 | The observed Poynting flux in the magnetosphere associated with the auroral arc. a** The encounter of RBSP-A and -B with the auroral arc during its azimuthal development in magnetic local time. **b** The frequency spectrogram of the magnetic field using the Morlet wavelet transform for RBSP-A. **c** The 3D wave Poynting flux (filtered from 0.6 mHz to 100 mHz) in a field-aligned coordinate (FAC) system, where the first component is along the background magnetic field (obtained by a running average over a 20 min window), the second component is along the azimuthally westward direction, and the third component completes the triad system (positive in the outward sense). **d**, **e** are for RBSP-B, in the same format as (**b**, **c**). Higher cadence Poynting flux (0.6 mHz to 4 Hz) is available at RBSP-B, showing a peak value around 350 mW/m². For each satellite, the vertical dotted lines in (**c**, **e**) mark the 2 minutes right after the dipolarization time in **a**. Detailed spectral information within the 2 minutes for the Poynting flux and E/B ratio are presented in the Fig. 5.

field measured at RBSP-A and -B in Fig. 2d shows a significant increase of the elevation angle (around 08:07:50 and 08:10 UT respectively), indicating the release of magnetic energy. To further investigate the energy route, we analyze the electric and magnetic fields in detail.

During the auroral arc crossing, strong electric and magnetic waves were simultaneously detected in the magnetosphere along the same magnetic field line, allowing us to quantitatively assess the electromagnetic energy flow above the auroral acceleration region. The observed waves are identified as Alfvén waves (**Methods, subsection Identification of kinetic Alfvén waves**), which play a key role in accelerating electrons along the magnetic field[10,15]. Figure 3a shows that RBSP-A and -B encountered the arc around 08:07:50 UT and 08:10 UT (vertical dashed lines), and that the Alfvén wave has a large component of the Poynting flux directed mainly Earthward (i.e., $S_{\parallel}$). The estimated energy in the waves was found to be sufficient for producing the observed auroral emissions (see Fig. 6). Detailed analysis of energy budget is provided in **Methods, subsections Calculation of Poynting flux, calculation of electron energy flux by potential drop, and estimation of the O+ energy flux**.

### The energy conversion below the AAR
The precipitating electrons shown in Fig. 1a are quantitively analyzed in Fig. 6. The integrated electron energy flux (i.e., having a peak energy flux about 162 mW/m²) is generally consistent with the result derived from the auroral emission itself (i.e., a peak energy flux about 87 mW/m²), indicating that the auroral particle acceleration region above the DMSP satellite was sufficient to provide all power for the aurora. Moreover, the majority of electron energy flux was contributed by the inverted-V structure, providing further evidence that the potential drop is the main process in converting the electromagnetic

energy of the Alfvén wave to the auroral precipitating particles. To efficiently accelerate electrons towards the Earth, the electric field associated with the potential drop should be directed upward, which would thus accelerate ionospheric oxygen ions (O+) to escape from the Earth. The ionospheric O+ ions would experience different fractions of the potential drop, thus exhibiting an energy dispersion when remotely observed[16–18]. Based on the O+ dispersion feature, we estimate that the potential drop acceleration operated around 1.5 $R_E$ altitude (**Methods, subsection Estimation of the AAR altitude**).

The strong correlation among the aurora, precipitating electrons, and escaping O+ reveals that the auroral acceleration region should be located around 1.5 $R_E$ above the ionosphere, and the potential drop is the final step needed to accelerate electrons for producing the auroral arc. To further assess the role of Alfvén waves in producing the potential drop, we investigate the electromagnetic energy flow, i.e., the Poynting flux, measured by the two RBSP satellites during the traversal of the arc. The Earthward Poynting flux is about 60 mW/m² as calculated from the spin fit electric field (up to about 100 mHz) at both satellites (red lines in Figs. 3c, e). However, comparing the frequency spectrogram of the magnetic field at RBSP-A (Fig. 3b) to that at RBSP-B (Fig. 3d), we note that the wave power at the smaller kinetic scales does exist at both RBSP-A and -B (>100 mHz). The availability of higher cadence electric fields at RBSP-B allows us to include this power, resulting in a better estimate of the actual Earthward Poynting flux (Fig. 3e orange line). The estimated Poynting flux of 350 mW/m² (see also Fig. 5c) is enough to account for the estimated kinetic energy fluxes of the inverted-V electron (162 mW/m²) and the escaping O+ (44 mW/m²). The estimated energy efficiency from the wave to accelerate electrons and O+ is thus 59%, suggesting that the energy conversion within the potential drop is likely to be high.

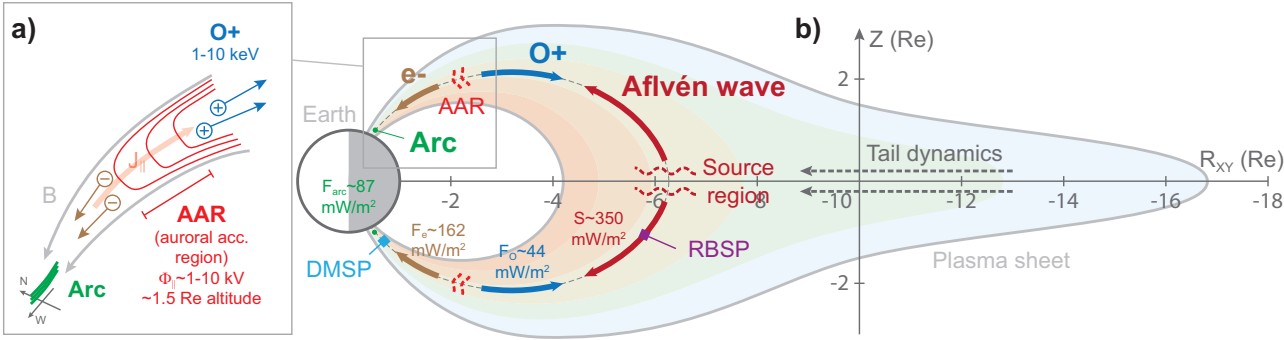

**Fig. 4 | Illustration of the coupling between magnetospheric processes and the auroral acceleration region. a** Details of the auroral acceleration region (AAR) inferred from observations with the well-known behavior of the associated upward parallel electrical current ($J_\parallel$) and charged particles. Electrons are accelerated toward the ionosphere, producing discrete arcs. O+ ions are accelerated away from the Earth. Our electron and ion observations suggest that the parallel potential drop $\Phi_\parallel$ of the AAR was 1–10 kV and around 1.5 $R_E$ above the ionosphere (Figs. 6 and 7). **b** Summarization of the energy fluxes along the magnetic field lines of the AAR, the nominal values of these energy fluxes, and other relevant processes. The detailed analysis of the energy budget is shown in Fig. 6 (for the discrete arc $F_{arc}$ and inverted-V electrons $F_e$), Fig. 7 (for the O+ outflows $F_O$), and Fig. 3 (for the Poynting flux $S$). The energy fluxes obtained from our observations suggest that the Alfvén waves, presumably generated by dynamical processes in the magnetotail, power the discrete arc via the electron and ion acceleration within the AAR.

Figure 4 summarizes the energy processes measured in the magnetosphere, auroral acceleration region, and precipitation. Our observations suggest that the source region that ultimately powers the AAR locates in near-Earth magnetotail, converting free energy to Alfvén waves (i.e., shear mode waves), which efficiently transmit electromagnetic energy along the magnetic fields toward the polar regions. A significant portion of the energy would be converted within the potential drop that can accelerate electrons to form the often-observed inverted-V structure. In this study, the auroral electrons are mainly energized by the potential drop, rather than directly from the parallel electric field of Alfvén waves (either the kinetic or inertial Alfvén waves[19]).

The potential drop above the Earth is quasi-static on the timescale of the electrons' transit time, which is on the order of seconds. This quasi-static nature invoked many theories based on electrostatic solutions to the Vlasov-Poisson equations including double layer and electrostatic shocks[20,21]. Although such electrostatic structures could explain many features of the potential drop, how the steady state is reached requires electromagnetic treatments. For example, Alfvén waves could develop parallel potential drop of several kV during the wave propagation toward the ionosphere[22–24]. Alfvén waves could also couple to the ionosphere and thus develop potential drops due to turbulence[25], anomalous resistance[26], and wave-wave mode conversion[27]. Our observations favor an electromagnetic theory as the Alfvén waves are closely related to the electron and ion acceleration through the potential drop. According to our observation, the main wave energy, carried by the "regular" Alfvén waves at magnetohydrodynamic (MHD) scales, is transported along the field lines towards the ionosphere. The exact mechanism on how Alfvén waves are dissipated through the potential drop is the remaining question for future investigations.

The Alfvén waves that we observed were well within the inner-magnetosphere, where the Earth's dipole field dominates. Downstream of the solar wind, the Earth's field lines are stretched into a tail-configuration, within which localized dynamical processes such as bursty bulk flows (BBFs) transport free energy toward the Earth[28–31]. A BBF contains localized fast flows that excite shear-mode waves around the sides due to flow shears and compressional waves around the Earthward head. We propose that the observed Alfvén waves could arise from either the shear mode waves of BBFs or the mode-conversion from the compressional mode waves[32,33] of BBFs. As these Alfvén waves tend to be more intense around density gradients[34], it seems to favor the generation through mode-conversion[22].

Potential drops were found to exist at both Earth and Jupiter, suggesting a common energy process across planets with global magnetic fields. It is natural to expect that similar processes would exist at other magnetized planets such as Saturn, considering many similar auroral dynamics amongst these planets[35–41]. Nevertheless, the potential drop magnitude at Jupiter typically exceeds that of Earth by one order of magnitude. Although it is still a mystery on the controller of the magnitude, we can reasonably speculate that the planetary magnetic field strength plays a key role. The magnetic field at auroral source region is 7-8 Gauss at Jupiter[42,43], which is about 20 times greater than the one at Earth. The value is quite consistent with the energy difference, providing a hint on the possible controller, which is yet to be understood. Although electric potential drop is an important step for magnetospheric energy conversion to auroral particles, we shall also point out that auroral electrons may also be directly accelerated by Alfvén waves[44,45], and auroral emissions are found in varied types of celestial objects, such as comet with no magnetic field[46], Mars with crustal field[47,48] and Jupiter's moons[49].

Using a fortuitous set of coincident observing platforms measuring different facets of the Earth auroral event on April 16, 2015, we provide strong evidence that the electromagnetic energy of the Alfvén waves powers the Earth's auroral arc through a static potential drop that converts this energy to kinetic energy of inverted-V electrons and O+ ion outflows. Our findings underscore the interplay between the Alfvén waves and the potential drop and provide a comprehensive view of how auroral arcs form on Earth and potentially other planets and astrophysical objects.

## Methods

### Satellites and instrumentation

The Van Allen Probes, also known as the Radiation Belt Storm Probes (RBSP-A and -B), contain two identical satellites in equatorial orbits around the Earth. The orbital apogee is 5.8 $R_E$ (Earth Radii) and thus is well above the auroral acceleration region (AAR), which is typically 1-3 $R_E$ above the ionosphere[50]. Data used include the electric field, magnetic field, thermal plasma measurements from the Electic Field and Waves instrument (EFW)[51,52], the Electric and Magnetic Field Instrument Suite and Integrated Science (EMFISIS)[53,54], and Helium Oxygen Proton Electron instrument (HOPE)[55]. Routinely, EMFISIS provides 3D magnetic field $\vec{B}$ at 64 samples/sec and EFW provides 2D electric field at 32 samples/sec within the plane that is perpendicular to the spin-axis of the satellites. The spin-axis electric field is calculated based on the $\vec{E} \cdot \vec{B}_0 = 0$ assumption from the ideal magnetohydrodynamic (MHD) condition. Here $\vec{E}$ is the 3D electric field in the rest frame of the plasma corotating with the Earth and $\vec{B}_0$ is the background magnetic field obtained by low-pass filtering $\vec{B}$ over a 20 min window (0.83 mHz).

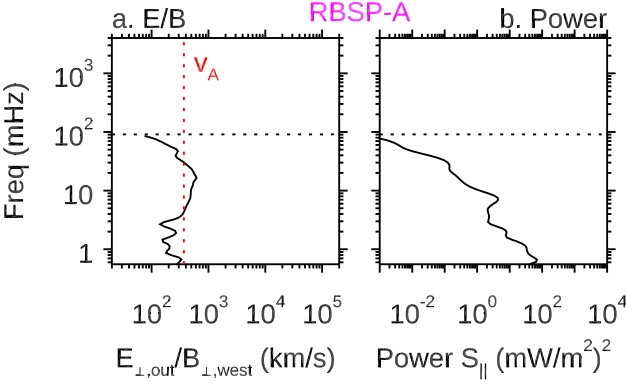
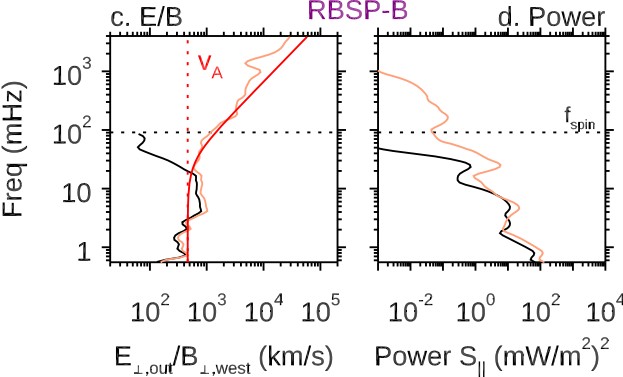

**Fig. 5 | Spectral analysis of the electric and magnetic fields at RBSP-A and -B.**
**a** The E/B ratio as a function of frequency in the spacecraft frame. The vertical dotted line marks the local Alfvén wave speed $v_A$. **b** The power of the parallel Poynting flux $S_\parallel$. **a**, **b** are RBSP-A results for the spinfit resolution data. **c**, **d** are for RBSP-B in the same format as (**a**, **b**). In addition, the orange curves in (**c**, **d**) are for the E/B ratio and power of $S_\parallel$ for the high-resolution data. The red curve is the dispersion relation of kinetic Alfvén waves (KAWs). In all panels, the horizontal dotted line marks the frequency corresponds to the spacecraft's spin $f_{spin}$. The quantities are averaged over the 2 minutes interval marked in Figs. 3c, e, coinciding with the largest Poynting flux at each satellite.

HOPE measures the thermal plasma from several eV to 50 keV per charge for electron, H + , O + , and He + , with a cadence of 22 sec and a pitch angle resolution of 18 deg.

EFW provides, in addition to the survey resolution (32 samples/sec) electric field, a spin fit electric field at the cadence of one sample per spin period of the spacecraft (about 11 sec)[52]. The survey resolution electric field is available when all four EFW spin plane probes are working properly, whereas the spin fit electric field is available when at least two of them work. In this event, only RBSP-B had the survey resolution electric field, yet both RBSP-A and -B have the spin fit electric field.

The Defense Meteorological Satellite Program (DMSP) F19 satellite was in a polar orbit around the altitude of 840 km. Data from the Special Sensor UV Spectrographic Imager (SSUSI)[56] and the Special Sensor J (SSJ)[57] are used to provide a snapshot of the auroral arc in the southern hemisphere and the associated inverted-V electrons respectively. Around the magnetic conjugate location of the auroral arc in the northern hemisphere, the ground all-sky imagers (ASIs) in northern America[58] provide further details on the spatial and temporal evolution of the arc at the resolution of 1-5 km and 3 sec respectively

### Coordinate

The Solar Magnetospheric (SM) system is commonly used within the portion of the geospace dominated by the Earth's dipole field. The z-axis is exactly aligned with the Earth's dipole axis. The y-axis is perpendicular to the Sun-Earth line and z-axis. The x-axis completes the triad system. In addition, the field-aligned coordinate (FAC) is used to present the electric and magnetic fields and Poynting flux. In this coordinate system, the b-axis is along the background magnetic field $\vec{B}_0$; the w-axis points azimuthally westward; the o-axis points poloidal outward.

### The identification of kinetic Alfvén waves (KAWs)

Intensive low-frequency electromagnetic waves like those shown in Fig. 3 have been routinely observed in the Earth's plasma sheet and its boundary layer during geomagnetic active times[11,34,59]. Such waves often have a broadband frequency spectrogram over ion gyrofrequencies, suggesting that they are not cyclotron waves, and that the broadband nature is a result of Doppler shift[60]. Considering the Doppler shift due to the relative velocity $\vec{v}_f$ between the spacecraft and local plasma, the dispersion relation for Alfvén waves in kinetic theory is $\omega/k_\parallel = v_A\sqrt{1+k_\perp^2\rho_i^2}$. Here $\omega$ is the angular frequency, $k_\parallel$ and $k_\perp$ are the parallel and perpendicular wave vectors, respectively, $v_A$ is the

Alfvén speed, and $\rho_i = \sqrt{k_B T_e/m_i}/\omega_{ci}$ is the ion acoustic gyro-radius related to the electron temperature $T_e$, ion mass $m_i$ and ion cyclotron angular frequency $\omega_{ci}$. This dispersion relation describes the kinetic Aflven waves, for plasma beta $\beta \in (m_e/m_i, 1)$[15,61], where $m_e$ and $m_i$ are the electron and ion masses, respectively.

The parallel phase speed of kinetic Alfvén waves can be approximated in terms of the wave frequency in the spacecraft frame $f_{SC}$[60,62,63] as

$$\frac{E_{\perp, out}}{B_{\perp, west}} \simeq v_A\sqrt{1+\frac{f_{SC}^2 \rho_i^2}{v_f^2}} \qquad (1)$$

Here, since the local plasma contained both H+ and O + , we used the averaged ion mass $m_i = (n_H m_H + n_O m_O)/(n_H + n_O)$ and the associated ion cyclotron angular frequency $\omega_{ci} = eB/m_i$. $E_{\perp, out}$ and $B_{\perp, west}$ are the electric field along the o-axis and the magnetic field along the w-axis, respectively. The axes are defined in **Methods, subsection Coordinate**. The flow speed $v_f$ is taken to be 42 km/s, which is scaled from the angular speed of the azimuthal development of the auroral arc shown in Fig. 3a. Given the arc developed primarily azimuthally without significant change in MLat, we infer that the azimuthal flow is the dominant component[64,65].

Figure 5 shows the comparison between the measured E/B ratio and the wave dispersion relationship as a function of $f_{SC}$. The quantities are averaged over the two minutes interval marked in Fig. 3c and e, coinciding with the largest Poynting flux. For RBSP-A, since only the spin fit electric field is available, the comparison is limited to $f_{SC}$ below 100 mHz. Within this frequency range, we can see that the measured E/B ratio (black) is consistent with the local Alfvén speed (red) derived from other independent measurements (Fig. 5a). This and the same comparison at RBSP-B (Fig. 5c) suggest that these low-frequency waves ($f_{SC} < 100$ mHz) are "regular" Alfvén waves (shear mode at MHD scales). Kinetic theory shows that Alfvén waves exist as the inertial Alfvén waves for $\beta \ll m_e/m_i$ and the kinetic Alfvén waves for $m_e/m_i \ll \beta \ll 1$[15]. The difference between the two modes is obvious at kinetic scales, which Doppler shift to high frequencies in the spacecraft frame. At RBSP-B, which has higher resolution electric field measurements, the behavior of the Alfvén waves at kinetic scales is consistent with the dispersion relationship of the kinetic Alfvén wave shown in Eq. 1 (orange lines in Fig. 5c), suggesting that the Alfvén waves observed at RBSP -B are kinetic Alfvén waves. Given the similar high-frequency waves (>100 mHz) as seen in the magnetic field (Figs. 3b, d), we infer that RBSP-A also observed the

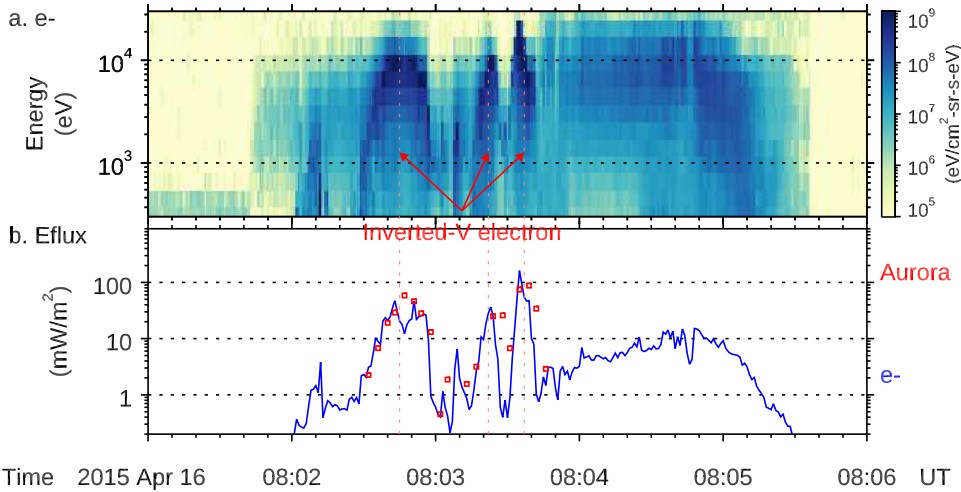

**Fig. 6 | Derivation of energy fluxes of the auroral arc and the inverted-V electrons. a** The electron energy-time spectrogram measured by the DMSP F19 satellite. **b** The electron energy flux (blue) integrated from the energy-time spectrograms and the energy flux of the auroral arc (red dots) along the DSMP F19's footprint (c.f. Fig. 2b).

"regular" and kinetic Alfvén waves and with similar magnitude to RBSP-B. Figure 5b and d show that the power of the Poynting flux associated with the Alfvén waves peaks around the 10 mHz frequency range.

We note that in this paper, the term Alfvén wave refers to the shear-mode Alfvén waves including the wave power at both the MHD and kinetic scales, which correspond to the "regular" and kinetic Alfvén waves, respectively.

### The calculation of poynting flux

The "Alfvénic Poynting flux", which is the Poynting flux associated with Alfvén waves, has been extensively studied in the past several decades in the Earth's magnetosphere in terms of observations[9,34,59] and simulations[66,67] and has been recently studied on Jupiter[68]. To resolve the frequency dependence of the Poynting flux, we adopt the following equation to calculate the Poynting flux spectrogram as a function of time $t$ and frequency in the spacecraft frame $f_{SC}$

$$\vec{S}(f_{SC}, t) = \vec{E}(f_{SC}, t) \times \vec{B}(f_{SC}, t),$$

where $\vec{E}(f_{SC}, t)$ and $\vec{B}(f_{SC}, t)$ are the real part of the Morlet wavelet transformation. The vector Poynting flux is integrated over the covered frequency

$$\vec{S}(t) = \int_{f_l}^{f_h} df_{SC} \vec{S}(f_{SC}, t)$$

This is the Poynting flux presented in Figs. 3c, e. The upper frequency limit $f_h$ is usually limited by the cadence of the electric and magnetic fields. The lower frequency limit $f_l$ is usually determined by the duration of the significant Poynting flux. To best estimate the total Poynting flux, the low and high frequency limits ($f_l$ and $f_h$) are chosen to enclose the main power in $\vec{S}(f_{SC}, t)$. This procedure has been documented and used in many previous papers[34,62,69]. A typical choice of the low frequency limit is 5.5 mHz[9,70], which captures the main Poynting flux power in general. Here, the low frequency limits $f_l$ is chosen to be 0.6 mHz, which returns a slightly larger (2%) maximum Poynting flux. Here we estimate the uncertainty of the Poynting flux $\vec{S}(t)$ is on the order of 2%, related to the lower limit to be used.

### Calculation of electron energy flux by potential drop

DMSP electron measurements do not cover all solid angles. Therefore, estimating the energy flux associated with the inverted-V

electrons depends on assumptions of the electron distribution. Typical inverted-V electrons are isotropic in pitch angle except the loss cone[71]. In this study, we assume such a distribution to estimate the electron energy flux. The results are shown as the blue line in Fig. 6b. Independently, we estimate the auroral energy flux based on the DMSP SSUSI snapshot. The results are overplotted as red dots in Fig. 6b. The energy flux of the inverted-V electrons correlates well with that of the auroral arc. This is well expected as the inverted-V electrons are the direct energy source for exciting the ionospheric atoms which emit the auroral light.

DMSP F19 crossed the auroral arc of interest that is shown in Fig. 2b. The arc corresponded to the third inverted-V electron structure in Fig. 6b. The peak electron energy flux is 162 mW/m². In this paper, to compare energy fluxes at different altitudes, they are normalized to 100 km altitude according to the scaling of a constant magnetic flux. The peak auroral energy flux is 87 mW/m², which is 54% of the estimated electron energy flux. The peak Poynting flux at RBSP is estimated to be 350 mW/m² (Fig. 3e). In Methods, subsection Estimation of the O+ energy flux, we estimate the kinetic energy of the O+ outflows to be 44 mW/m². The energy conversion rate of the potential drop from the electromagnetic energy of the Alfvén waves to the kinetic energy of the inverted-V electrons is 46%. The rate is 59% if O+ outflows are also included. Hence, the energy conversions from the Poynting flux to the inverted-V electrons and then to auroral emissions are both highly efficient.

### Estimation of the AAR altitude

O+ outflows with clear energy-time dispersions are routinely observed during strong auroral activities on Earth[18,72]. Such dispersion is also observed in this event, as shown in Fig. 7d. The angle between the O+ velocity and the background magnetic field, which is called the pitch angle, was within 30 deg (Fig. 7c). The dispersion is caused by a time-of-flight effect of O+ at different energies. Hence, tracing these O+ backward in time and space along the magnetic field line allows us to determine when and where they originated. The tracing results are shown in Fig. 7. The test particles for the tracing are marked as orange dots in Fig. 7d and are selected as follows: for each time tag, the test particle's energy corresponds to the maximum flux. Figure 7e shows the tracing results, where each orange line is the time history of the motion of a test particle along the field line. The crossing region of the history traces is determined to be around 2.5 $R_E$ geocentric distances (1.5 $R_E$ altitude), corresponding to the location of minimum spread (standard deviation) of the tracing lines

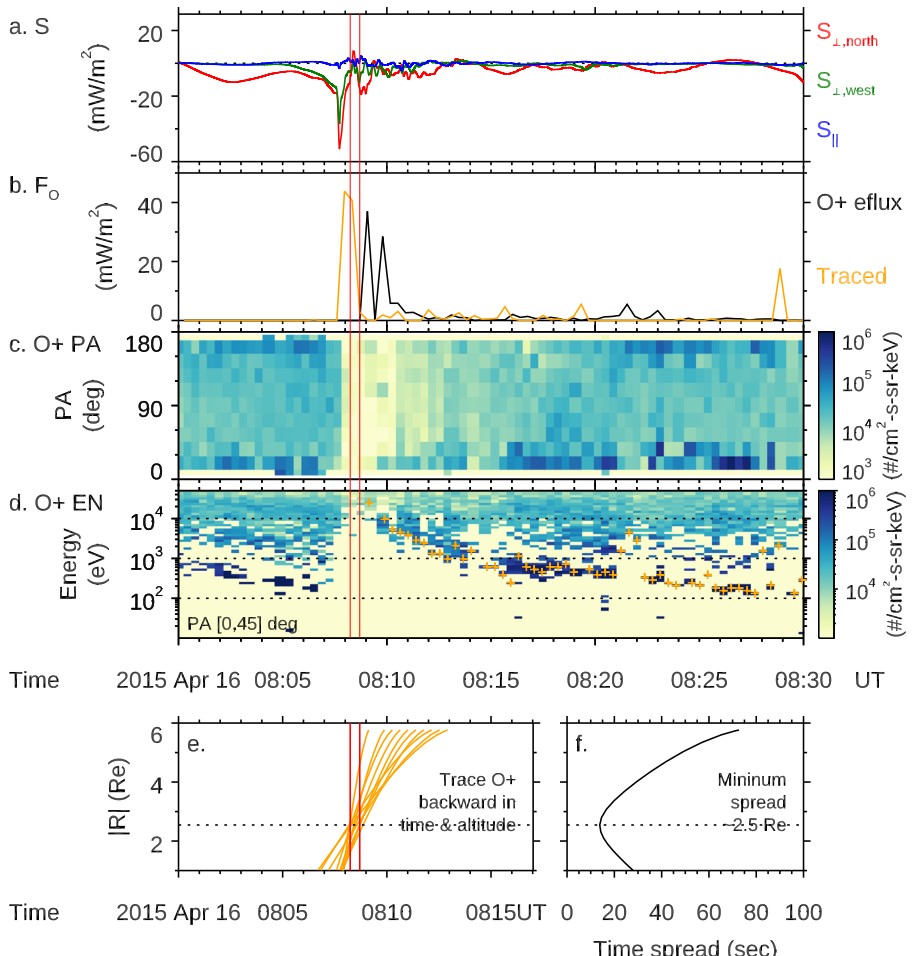

**Fig. 7 | Derivation of the O+ outflows' kinetic energy flux (F$_O$) and the originated time and altitude. a** The Poynting flux due to Alfvén waves. **b** The measured O+ energy flux (black) and that after tracing back in time and altitude (yellow). **c**, **d** The pitch angle and energy spectrogram for the O+ ions. Test particles for tracing are marked by yellow signs in (**d**). **e**, **f** Details of the tracing for 1–10 keV O+ ions. The time-altitude trajectories of the test particles converge around 2.5 R$_E$ geocentric distances and around 08:08:30 UT.

(Fig. 7f). As previous statistical studies have shown that the typical altitude of the potential drop is 2-4 R$_E$ geocentric distances (Karlsson et al.[52] and references therein), our tracing result is consistent with typical expectations. In addition, our tracing determines the time when these O+ are generated to be around 08:08:14-08:09:42 UT (marked by the vertical red lines). This time is about 30 sec after the peak of the Earthward Poynting flux of the Alfvén waves (Fig. 7a). The time lag may be related to the travel time of Alfvén waves to the ionosphere (typically 10–15 sec) and the travel time of O+ from the ionosphere to and through the potential drop.

Note that O+ outflows can be generated by two mechanisms. One is through parallel acceleration of a potential drop (0 deg or 180 pitch angle) and the other is through perpendicular heating (90 deg pitch angle) of a variety of waves[73–75]. The perpendicularly heated ions will be elevated in altitude by the mirror force due to the dipole configuration and hence folded toward 0 deg pitch angle (or 180 deg) as their altitude increases. At RBSP, the folding is so significant that the pitch angle distributions of parallel accelerated and perpendicularly heated O+ cannot be clearly distinguished. However, with the DMSP inverted-V electron observations, we infer that the O+ outflows of 1–10 keV are due to parallel accelerations because two reasons: Firstly, the inverted-V electrons are 1–10 keV, suggesting the existence of a potential drop of 1–10 kV; Secondly, the timing of the O+ outflows is consistent with a parallel acceleration, (c.f. Fig. 7a, b).

### Estimation of the O+ energy flux

We calculate the energy flux of the O+ outflow using the test particles determined in the previous subsection. We assume that the distribution of O+ has a spread of 3 energy bins around the energy of the test particles and that the distribution is uniform in [0, 30] deg pitch angle. In Fig. 7, the integrated energy flux is normalized to 100 km altitude (black line in Fig. 7b). To further correct for the energy-time dispersion, we trace this energy flux backward in time and obtain the traced energy flux (orange line in Fig. 7b). The peak value of the traced energy flux is 44 mW/m$^2$.

### Data availability

The Juno-JEDI data is available via (https://pds-ppi.igpp.ucla.edu/collection/JNO-J-JED-3-CDR-V1.0:DATA). The DMSP SSJ data is available via (https://cedar.openmadrigal.org/) through publicly accessible data requests. The RBSP data is publicly available at (https://cdaweb.gsfc.nasa.gov/pub/data/rbsp/), in which data from the relevant instruments can be found under the corresponding subdirectories. The source data are provided in (https://zenodo.org/records/17421615) with the identifier (https://doi.org/10.5281/zenodo.17421615).

### Code availability

The codes used in this study are available from the corresponding authors upon request. Our codes are essentially combinations of the

routines provided by the SPEDAS software package (http://spedas.org/wiki/index.php?title=Main_Page), to read the relevant spacecraft data, perform basic computations explained in **Methods**, and to generate the figures.

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

## Acknowledgements

Z.Y. acknowledges the General Program of the National Natural Science Foundation of China (Grant 42374212), the Research Grants Council (RGC) General Research Fund (Grant No. 17309124) and Project (JLFS/P-702/24) of Hong Kong RGC Co-funding Mechanism on Joint Laboratories with the Chinese Academy. S.T. acknowledges NASA grants 80NSSC22K1027 and 80NSSC19K0306. Y.S. acknowledges NASA fund 80NSSC25K7673. S.T. thank Dr.Y.S. and Prof.C.C. for the discussions and help on the related works.

## Author contributions

S.T. and Z.Y. wrote the manuscript. S.T. and R.W. conducted the research work. R.L.L., J.B., and L.R.L. assisted with the data interpretation. J.L. and R.S. assisted with data-simulation comparisons. CPF, YS, and GDR assisted with data analysis.

## Competing interests

The authors declare no competing interests.
