## [Transparent Peer Review file · Nature Communications]

Evidence for Alfvén waves powering auroral arc via a static electric potential drop

Corresponding Author: Dr Sheng Tian

Version 0:

Reviewer comments:

Reviewer #1

(Remarks to the Author)

Review of "The energy source powering aurora through a static electric potential" by S. Tian and coworkers.

The authors present a case study based on multi-point observations made on the surface of the Earth and in surrounding space that shed light on how the magnetosphere is coupled to the ionosphere. Specifically, they show how kinetic Alfvén waves launched from the magnetotail lead to the formation of a quasi-static magnetic-field-aligned electric potential above the ionosphere that accelerates particles down into the atmosphere to produce an observed auroral arc.

I recommend this paper be rejected, with re-submission encouraged. The manuscript needs a significant amount of work, but I think the authors should be prompted to try again if they think they can better explain the significance of the key result they present.

Major, overall comment: The context and importance of the results presented is not clear to me as a space physicist who is not an expert on this specific topic, and it certainly will not be clear to the reader. What exactly is mysterious about energy dynamics associated with the auroral acceleration region? What is the reason this mystery has persisted, despite many observations? Are there competing ideas to explain whatever the specific problem is? What is it about the observations presented here that allows the present authors to do better? Is it really surprising that changes in the magnetosphere are caused by Alfvén waves propagating through the system? Why does resolving the proposed mystery matter, what are the implications for this field and beyond?

I suspect these questions may be answerable (giving the authors the benefit of the doubt). Combining this with the current absence of clear answers in the text is the basis of my recommendation to reject, encouraging re-submission.

Abstract: There is very little content here. See above comments. Most of the questions I list should have been answered by the end of reading a revised version of the abstract.

Why include Jupiter at all? Seems done for the sake of it, without a clear rationale. This physics likely happens in some form at every planet, in the Solar System and beyond, so why only show some data from Juno at Jupiter? I suggest limiting the scope of the manuscript to the Earth system, highlighting implications for other systems in the discussion.

Figure 1: Poor quality, needs work to improve clarity of axes labels, etc.

Reviewer #2

(Remarks to the Author)

Review of manuscript "The energy source powering aurora through a static electric potential" by S. Tian et al.

From an auroral event on 16 April 2015, the authors evaluate data simultaneously obtained with four different means at three different altitudes in a unique constellation between satellites and aurora. The data covered electric and magnetic fields measured by the two radiation Belt Storm Probes, RBSPs, near 5.8 RE, electron fluxes measured by the Defence Meteorological Satellite Program, DMSP, at 840 km height, and observations of all sky cameras on the ground. The authors focus on the energy flow into the acceleration region of the auroral particles, differentiating between electrostatic and wave

acceleration. The paper is composed of a Main part containing the results and a Method part meticulously describing how the data were obtained and analysed. The central quantity to be derived was the Poynting flux and its comparison with the flux of the electrons and upward accelerated O⁺ ions. The main conclusion was that kinetic Alfvén waves, excited in the source region encountered by the RBSP spacecraft, were carrying the energy into the acceleration region between 2 and 4 RE altitude with sufficient power to cope for the observed particle fluxes.

The paper is most interesting. The separation into main results and methods is very beneficial for the reader and illuminating for the expert. All parameters and the instrumentation are well described. Much emphasis is placed on the identification of the waves, their spectra, and energy fluxes. A particularly nice piece of work is the determination of the height of the acceleration region from the dispersion of O⁺.

I have a few reservations about two conclusions. (1) The authors suggest that the Poynting flux carried by kinetic Alfvén waves is absorbed in the auroral acceleration region (lines 117/118, 142/143, 171/172, 535/536), admitting, however, that they are unable to propose a mechanism. It is commonly thought that kinetic or inertial Alfvén waves in the magnetosphere have frequencies higher than shown in Figure 3 and are absorbed at low heights generating strongly field-aligned electrons with a diffuse energy spectrum. By contrast, the Poynting flux derived from the RBSP wave data is dominated by the low frequencies. These are most likely an expression of the plasma dynamics at the source region, as indicated by the strong changes in tilt angle of the magnetic field (Figure 2d). The main effect of the agitated plasma is bending or shearing of the magnetic field, which is transported along the field lines towards the ionosphere by regular, not kinetic Alfvén waves. On the other hand, the concentration of the wave energy flow into downward and westward direction is surprising. Some comments on that may be helpful. With respect to the interaction with the AAR, I find the paragraph from lines 176 to 187 not very illuminating on this subject.

(2) The derivation of the relative velocity of spacecraft and plasma from Figure 3a is rather questionable. The narrow, elongated shape of the arc suggests that it is generated from a boundary. The variation of the brightness along this boundary likely reflects variations of the agitation from higher altitude, as for instance by a BBF. However, I do not see how much this affects the conclusions.

Overall, a good paper, worth to study. Can be published essentially as is, after response to my comments.

Gerhard Haerendel

Reviewer #3

(Remarks to the Author)

The paper "The energy source powering aurora through a static electric potential" addresses the energy transport / balance in the auroral magnetosphere-ionosphere (M-I) by presenting an event study based on Van Allen / RBSP, DMSP F19, and ground THEMIS optical data. While the fundamental components of the energetics involved are understood nowadays rather well, close quantitative estimates are scarce, if at all. In this respect, the manuscript fills a significant empty spot and my overall assessment is that, essentially, the paper is suitable for publication in Nature Communications. However, I think the message can be better streamlined and the contribution made by the paper better pinpointed, as detailed below.

1. The authors emphasize the 'mystery' around auroral acceleration and particle energization, related to the relationship between quasi-static and Alfvénic acceleration (L27, 39, 45, 50, 203), as well as the 'discovery' made. I feel a bit hesitant on using those words, since the matter has been broadly addressed in the literature - as proven also by the rich selection of cited papers. The fundamental building blocks were summarized in several monographs (e.g., *Auroral Plasma Physics*, Springer, 2003, <https://doi.org/10.1007/978-94-007-1086-3>) and quantitative evidence for the conversion of the Alfvénic Poynting flux into particle energy flux was brought, e.g., in statistical studies based on Polar and FAST data (Keiling et al., 2002, <https://doi.org/10.1029/2001JA900140>, Chaston et al., 2002, <https://doi.org/10.1029/2002JA009272>, Chaston, 2006; in particular Figure 2, <https://doi.org/10.1029/169GM16>). These being said, I am not aware of an event oriented, careful evaluation of the auroral M-I coupling energetics, as performed in the paper - including Poynting flux above the acceleration region, precipitating electrons, auroral emissions, and outflowing ions. To my understanding, the importance of the paper resides mainly in this thorough quantitative analysis.

2. The authors emphasize as well the universal nature of auroral acceleration, which plays an important role in various environments, within and beyond the solar system. Similar to the processes responsible for auroral acceleration, this matter was addressed too in the past (e.g., AGU Monograph 197, *Auroral Phenomenology and Magnetospheric Processes: Earth and Other Planets*, <https://doi.org/10.1029/GM197>), and the importance of the parallel electric field (Alfvén, 1958, <https://onlinelibrary.wiley.com/doi/abs/10.1111/j.2153-3490.1958.tb01991.x>) was emphasized also for astrophysics (e.g., Haerendel, 1994, <https://adsabs.harvard.edu/pdf/1994apjs...90..765h>). On the other hand, the evidence in the paper builds on the event of April 16, 2015, observed at the Earth by conjugate satellite and ground data. This should be made clear upfront. Unlike the Earth panel of Figure 1, which is detailed and developed in the paper (including Extended data Figure 2), there is no follow-up to the Jupiter panel of Figure 1. Merging the Earth panel of Figure 1 with Figure 2 may help the streamlining, while the Jupiter panel could be added as a small inset, for comparison (or eliminated).

3. The authors refer to the 'arc' geometry of the aurora both in Figures 2-a ('Arc north'), 2-b ('Arc south'), in the caption of Figure 3-a ('... the auroral arc during its azimuthal development'), and in Figures 4-a, 4-b:

3a. The arc geometry is rather clearly visible in the DMSP F19 SSUSI data of the southern aurora, but less clear in the THEMIS ASI ground data of the northern aurora - even though, the 'skeleton' of the aurora looks, indeed, rather similar. Perhaps this skeleton can be better emphasized in Figures 2-a and 2-b, e.g., by adding some zoom-in sub-panels?

3b. The arc development of 4 deg/min is hard to follow in Figure 3-a. Please improve. It would be nice to add a panel similar to 3-a for the southern aurora, based on F19 SSUSI data - to pinpoint, perhaps, the similar dynamics in the two hemispheres (related to the similar skeleton under 3a?).

3c. I do not understand the negative ASI counts in the color scales of Figures 2-a and 3-a (which does not seem to matter to

the figures, since all the counts are bluish / positive). Perhaps the negative counts are related to outflowing energy fluxes (similar to the energy flux calibration of the SSUSI data?), but if so, this should be explained. In terms of ASI raw data, I guess the counts can only be positive.

4. The dynamics and geometry of the observed auroral event deserve more attention. The energy fluxes of 300+ mW/m² (Poynting) and 100+ mW/m² (electrons) are remarkably high, compare also with Figure 2 of Chaston, 2006 (see point 1). Likewise, the energy flux of 87 mW/m² is well above the visibility threshold (empirically ~1 mW/m²). Such energy fluxes are characteristic for the most dynamic auroras, which tend to develop as 2-D forms, rather than 1-D arcs. The association of high energy fluxes, specific to very dynamic events, with arc aurora, is quite remarkable and it might be insightful to see also the dynamics behind, as captured by the THEMIS ASI and DMSP F19 SSUSI (e.g., movie links to some open access repository?), in addition to the pdf still images. Indications on the geophysical indices (Kp, Dst, AE,...) would be helpful as well, to introduce the context of the event - a rather disturbed day, during an extended moderate storm.

5. Others

- L87 and Figure 2-c: The tilt angle, θ , is defined as the absolute value of $\sin^{-1}(B_z/B)$ (missing vertical bar, for absolute value, at the left side of this expression?). Based on Figure 2-c, where the tilt angle is rather small, between 20 and 40 deg, and on the satellites' location near the equatorial plane, the 'tilt angle' appears to be the angle made by the magnetic field with the z axis (equal to zero at the equator, in an ideal dipole configuration). If so, this angle should be $\cos^{-1}(B_z/B)$.

- L95-96: The tilt angle is now called elevation angle. Please keep to the same name, be it tilt or elevation. Apart from that, it is noted the increase at RBSP A and B, separated by a couple of minutes, but there is no comment on the simultaneous change, at ~08:07:50, from a slow decrease in both RBSP A and B, to a rapid increase in RBSP A and a rapid decrease in RBSP B (before the increase at 08:10). Any hint on what could be the reason behind? (perhaps in terms of BBF configuration, L192-196?) The increase of the tilt / elevation angle is associated with the release of magnetic energy. Is this an indication to (BBF related?) dipolarization? If yes, the tilt / elevation angle is rather made with the x-y (equatorial) plane (contrary to my inference above). Please clarify.

- L100: RBSP A and B sample the inner magnetosphere (as noted at L189), rather than the 'distant magnetosphere'. As a compromise, one could leave just 'magnetosphere'.

- L112, 114, 155, 156, 530-536 and Figure 4-b: The (maximum) energy fluxes for electrons and oxygen ions are indicated in the text as 162 and 44 mW/m², respectively, whereas Figure 4-b shows ~165 and ~50 mW/m². I understand the approximation, but it might be good to have the same numbers everywhere (also since the approximation is not applied to the 87 mW/m² of the auroral emissions). At L155 the energy flux of the escaping O⁺ is indicated wrongly as 87 mW/m². Should be changed to 44 and, accordingly, the efficiency at L156 should be $(162 + 44) / 350 = 59\%$, instead of 72 %. Same efficiency of 59% at L536, instead of 57%.

- L119, 120: In line with point 2, please replace 'planet' with 'Earth'. Certainly, there is ion outflow also at planets, but this very event is observed at Earth (and all the numbers characterize this event).

- L144-146: Please explain this consistency. See also point 3b above.

- L148-149: Not sure I understand this sentence. Do you mean that the Poynting flux is mainly contributed by frequencies under 100 mHz, which correspond to shear Alfvén wave regime, according to panels a) and c) of Extended data Figure 1 (E/B ratio around v_A)?

- L196-197: The relationship between most intense Alfvén waves and density gradients is not addressed in the paper.

- L483-484: Why this association, between the flow speed and the azimuthal development of the arc?

6. Typos and alike

L31: is launched from => resides in (?); L111: Figure 1b => 1a; L144: transverse => traversal (?); L145: crossingS; L146: shown IN; L168: powerS; L176: on Earth => above the Earth; L211: crystal => crustal; L324: Mhd => MHD (use {MHD} with bibtext, likewise for all capital letters / acronyms); L478: Full stop after ω_{ci} (?). Then: The velocity of kinetic Alfvén waves can be approximated... (?); L485: we infer this is => we infer that (?); L492: Panel a-1 => Panel a; L499: suggestING; L523, L524: Panel b-2 => Panel b of Extended data Figure 2; L525: correlated => correlates (?); L543: O⁺ => O⁺ velocity; L570: the timing ... IS consistent.

Version 1:

Reviewer comments:

Reviewer #1

(Remarks to the Author)

The authors have addressed all my questions. Although the manuscript presents solid, original findings, I am of the opinion that this is an incremental advance, without substantial implications for the field. Perhaps a different journal would be better fit, but I am not well qualified to judge.

Reviewer #2

(Remarks to the Author)

What are the noteworthy results? A concept of powering the auroral acceleration by kinetic Alfvén waves.

Will the work be of significance to the field and related fields? How does it compare to the established literature? If the work is not original, please provide relevant references. The work is original and will be significant through the outstanding data

set.

Does the work support the conclusions and claims, or is additional evidence needed? Yes, it does.

Are there any flaws in the data analysis, interpretation and conclusions? Do these prohibit publication or require revision? There are problematic interpretations. No revision required.

Is the methodology sound? Does the work meet the expected standards in your field? Yes.

Is there enough detail provided in the methods for the work to be reproduced? Yes.

2nd Review of manuscript "The energy source powering aurora through a static electric potential" by S. Tian et al.

I maintain my previous judgment that this is an interesting paper worth publishing. However, I now have some hesitation to add the words "as is". The point is that the authors did not react properly to my comment: "These are most likely an expression of the plasma dynamics at the source region, as indicated by the strong changes in tilt angle of the magnetic field (Figure 2d)." In their reply, the authors say after Figure 1: "Here, we examined the local plasma beta around the RBSP spacecraft and showed that it is in the beta range for the kinetic Alfvén wave ($1 > \beta > m_e/m_i$). However, the fact that the deviation of tilt angle from the background magnetic field is of the same magnitude as that field value suggests that there is a background plasma present with $\beta \sim 1$ and the condition for kinetic Alfvén waves: $\beta \ll 1$ is not fulfilled.

There are two conflicts, one about the proper value of beta and the other about the proper designation of the waves constituting the Poynting flux with peak at 10 mHz as kinetic Alfvén waves (Lines 516-518).

Conflict 1: The authors say in lines 520-521: "...the term kinetic Alfvén wave refers to the shear-mode Alfvén waves at $m_e/m_i \ll \beta \ll 1$ ".

Conflict 2: In lines 516-518: "Panels b and d show that the power of the Poynting flux associated with the kinetic Alfvén waves peaks around the 10 mHz frequency range".

I believe that the authors insist on the designation: "kinetic Alfvén waves", because they are thought to carry a finite E_{\parallel} , and that is needed, as "Energy carried by kinetic Alfvén waves travels from the magnetosphere to the auroral acceleration region and forms electric potential drop that further acceleration particles to produce aurorae." (Lines 30-32)

In my first review, I did not want to enter into a debate on the overall concept, which starts from an energy source in form of kinetic Alfvén waves and lets them be absorbed at lower altitudes to form an electrostatic potential drop. Now I see a need to do so.

I further question the validity of the formulation in lines 191-194: "According to our observation, the main wave energy is carried by the shearing of the magnetic fields, i.e., the "regular" Alfvén waves, transported along the field lines towards the ionosphere. The energy conversion around the AAR is likely "dispersive", i.e., through Alfvén waves at kinetic scales." Alfvén waves do not shear the magnetic field. The shear is in the magnetic field and caused by the field-aligned current. The Alfvén waves carry the energy gained by unshearing the magnetic field. This is in brief my understanding of the energization of the AAR:

Origin is a concentrated field-aligned current. It is connected with shearing the magnetic field and thus storing energy in the field. Next is the appearance of an electric resistance to the field-aligned current at heights above about 1500 km. It may be connected with the mirror effect according to (Knight 1973). The resistance causes the presence of a parallel potential drop, which decouples the magnetic field from the ionosphere. Decoupling means that the "field lines" (i.e. plasma and field) move in a way as to remove or lower the magnetic shear. This perturbation of the field propagates upward (and downward, but less important) with Alfvén speed constituting a front behind which the sheared field is unshaped. The released magnetic energy is transported by Alfvén waves towards the potential drop thus supplying it with the energy to be consumed by energization of electrons and ions.

I think that I owe the authors this summary to let them understand my above formulated criticism. However, I do not want them to modify their concept, just to take note of an alternate scenario and reply to my comments along lines 3 to 19.

Reviewer #3

(Remarks to the Author)

Thank you for the revision. While some issues have been fixed, a few require further attention:

1. The Response details the reasons for including the JUICE panel in Figure 1, in particular since Jupiter is the only other object apart from the Earth where inverted-V electrons were observed (I thought the same held for Saturn?) However, those reasons were not included in the text. Please clarify that also in the text, such that the readers can grasp easily the message of the JUICE panel, despite the fact that further analysis concentrates just on the event at Earth.

2. The authors indicate in the Response that "The several 100s mW/m² Poynting flux is very typical at the altitude above AAR. We have dozens of other similar events from RBSP and THEMIS." This comes somewhat at odds with former studies, like Keiling et al. (2002), in particular their Figure 5. The number of intense Poynting flux events there, well above 100 mW/m², seems rather limited, 4 out of 40, i. e., some 10%. Could this be an effect of the different orbit? Of the different data processing? Of your better statistics? Some other reason? Please comment.

3. Given the major importance of the quantitative argument for the paper, please include the comment above also in the text. Along the same line, regarding the data processing, your peak Poynting flux of 350 mW/m² relies on a frequency range whose lower limit is 0.6 mHz (L138, 143). This is different from Keiling et al. (2002), who subtract a background averaged over 3 min, roughly equivalent to a lower limit of 5.6 mHz, i.e., one order of magnitude larger.

3a) It would be nice to provide a brief motivation for your choice of 0.6 mHz (which is consistent with the 25 min duration of the time interval in Figure 3 - but why this duration?)

3b) The lower limit of the frequency range for the wavelet spectrograms in Figure 3 and for the spectra in Extended data Figure 1 is (a bit less than) 2 mHz, corresponding to some 10 min. Any particular reason for not showing the very low frequencies, from 0.6-2 mHz (corresponding to 10-25 min)?

3c) I presume that Extended data Figure 1 shows cuts through spectrograms (not shown) of E/B and of S_{parallel}, similar to the spectrograms of B_{West} in Figure 3. Please indicate the time instant(s) of these cuts (perhaps the time(s) of the maximum Poynting flux?)

3d) Much of the intense Poynting flux is contributed by the lower frequencies (as it is with many other signals - most of the power is related to / needed for the larger scales). The one order of magnitude difference in the lower frequency limit is presumably essential for the large number of events with several 100s mW/m² Poynting flux, mentioned in the Response.

4. I am aware that deriving the Poynting flux from experimental data is difficult and I understand that coming up with a precise recipe is not the main target of this paper. On the other hand, the quantitative coherence of the energy budget, as emphasized in Figure 4, would be difficult to achieve, without, at least, *some* accuracy of the recipe. An immediate example is the difference between the maximum Poynting flux based on spin fit electric field, 60 mW/m², as compared to that based on higher frequency electric field, 350 mW/m² (a larger value which, incidentally, relies also on the frequency range <100 mHz, as shown by Extended data Figure 1). Obviously, the spin fit Poynting flux cannot account for the observed particle energy fluxes. The good agreement between the (maximum) higher frequency Poynting flux and the other energy fluxes, inferred from particle and optical data, provides also a self-consistent validation for the Poynting flux recipe (with due regard to the error margin). A comment on this matter, in the text, would be welcomed.

5. Minor

L32: acceleration -> accelerates.

L113: The detailed analysis of energy budget is indeed provided in Methods, but the energy budget seems to fit better with Figure 4 than with Extended data Figure 2.

Version 2:

Reviewer comments:

Reviewer #2

(Remarks to the Author)

The authors have appropriately responded to my concerns about the nature of the waves and the interpretation of the low-frequency magnetic perturbations. Therefore, I can recommend the publication of the revised manuscript.

However, I would like to ad a small advice. Attribution of the auroral event to a dipolarization event asks for some supporting evidence. The term "dipolarization" pertains to the development of the magnetic field of a flow burst in the tail, whereas the two RBSP spacecraft are definitely inside the magnetosphere and would sense the effect of the impact of a flow burst. Are the measured low-frequency magnetic fields consistent with that impact?

Reviewer #3

(Remarks to the Author)

Happy to recommend publication.

Please be aware that the blue text at L533-567 shows irregular behavior and likewise the line numbers (on top of each other).

Version 3:

Reviewer comments:

Reviewer #2

(Remarks to the Author)

The key result of the study is: "Energy carried by Alfvén waves travels from the magnetosphere to the auroral acceleration region, forming an electric potential drop that accelerates particles to produce aurorae." However, the second acceleration mechanism, directly through kinetic Alfvén waves, has not received direct attention, although much of the electron data from the DMSP might well be the result of it. Instead, the Alfvén waves are just seen as e.m. energy carriers into the AAR. Quite appropriately, the question how the wave energy is converted into the electric potential structure is left open, the main issue being the comparisons of wave energy and the energy of the directly observed electrons.

The new element is the relation of the wave energy to a dipolarization event. The rebuttal to Reviewer #2 contains new ideas about that matter. However, they have not found direct entry into the manuscript. The sentence in Lines 207-210 : "The energy conversion around the AAR is likely "dispersive", i.e., through Alfvén waves at kinetic scales. The exact mechanism on how Alfvén waves are dissipated through the potential drop is the remaining question for future investigations." is perhaps more confusing than illuminating. The first sentence compares a process with plasma entities, whereas the second sentence suggests that this process might consist of the dissipation of the waves. Fortunately, it is left open.

Here, the key problem of the paper becomes evident. First, the waves are launched from high altitudes, possibly caused by the impact of a flow burst on the magnetosphere. Secondly, the waves are dissipated or converted in an electrostatic potential structure by an unknown process. Why not starting with the generation of an AAR by an excessive field-aligned current launched by the impact of a flow burst and the AAR triggering the inflow of energy stored in the magnetic field by the same process. Then it is irrelevant whether the Poynting flux consists of inertial MHD Alfvén waves. Much of this has already appeared in my third review.

This remark does not need a response of the authors. However, it may initiate a different formulation of the sentences in Lines 207-210.

The paper can be published as is.

Response to the Referees' comments

We thank the referees for their knowledgeable and constructive comments and suggestions, and we appreciate their reviews very much, which significantly help us to revise and improve the manuscript. Below are our point-by-point responses to the referees' remarks. The original referees' comments are quoted in *italics* and our responses are in **blue** for convenience. All requested revisions have been made according to the referees' suggestions.

Reviewer #1 (Remarks to the Author):

Review of "The energy source powering aurora through a static electric potential" by S. Tian and coworkers.

The authors present a case study based on multi-point observations made on the surface of the Earth and in surrounding space that shed light on how the magnetosphere is coupled to the ionosphere. Specifically, they show how kinetic Alfvén waves launched from the magnetotail lead to the formation of a quasi-static magnetic-field-aligned electric potential above the ionosphere that accelerates particles down into the atmosphere to produce an observed auroral arc.

I recommend this paper be rejected, with re-submission encouraged. The manuscript needs a significant amount of work, but I think the authors should be prompted to try again if they think they can better explain the significance of the key result they present.

Major, overall comment: The context and importance of the results presented is not clear to me as a space physicist who is not an expert on this specific topic, and it certainly will not be clear to the reader. What exactly is mysterious about energy dynamics associated with the auroral acceleration region? What is the reason this mystery has persisted, despite many observations? Are there competing ideas to explain whatever the specific problem is? What is it about the observations presented here that allows the present authors to do better? Is it really surprising that changes in the magnetosphere are caused by Alfvén waves propagating through the system? Why does resolving the proposed mystery matter, what are the implications for this field and beyond?

I suspect these questions may be answerable (giving the authors the benefit of the doubt). Combining this with the current absence of clear answers in the text is the basis of my recommendation to reject, encouraging re-submission.

Reply: Thanks for the comments on the presentation of results. Following your comments, we have significantly improved the presentation on the points below.

(1) What exactly is mysterious about energy dynamics associated with the auroral acceleration region.

In the past decades, the accelerated auroral electrons have been extensively observed by many satellites, particularly the FAST and DMSP satellites. Two types of typical electrons are identified, i.e., monoenergetic electrons with typical energies at a few keV to 10s keV (i.e. inverted-V electrons) and broadband electrons with energies spanning orders of magnitude (from 10s eV to 1 keV).

The two typical features are suggested to correspond to different mechanisms. The inverted-V electrons correspond to the acceleration by a quasi-static potential drop, whereas the broadband electrons correspond to the acceleration by the wave electric field by the so-called dispersive Alfvén waves, which include inertial and kinetic Alfvén waves. Although it is commonly accepted that the acceleration of the broadband electrons is powered by high-altitude Alfvén waves, it is still unclear what is the high-altitude energy source for the acceleration of the inverted-V electrons. Here, high-altitude means above the auroral acceleration region, which is typically located around 1-3 R_E above the Earth's ionosphere.

We have highlighted the description of the scientific question about auroral acceleration in the revised manuscript in abstract (e.g., line 26-27) and the main context (line 44-45).

(2) What is the reason this mystery has persisted, despite many observations?

Reply: Many observations have revealed key features related the auroral processes, such as the monoenergetic and broadband electron distributions that inform potential mechanisms of the acceleration. To reveal the whole chain of particle acceleration from the magnetosphere to auroral region during an active auroral event is, however, much more challenging. It is rather rare to have simultaneous observations from ground and space at low-altitude and high-altitude source region, when an active event occurs. In this study, an event was very luckily captured by satellite at all the key regions. Using the unprecedented measurements, we can calculate the precipitating Alfvénic energy, particle energy, location of energy conversion region, and the intensities of aurora and ground magnetic perturbations, allowing us to directly assess the energy budget in the energy conversion processes, thus, to reveal their relation.

Are there competing ideas to explain whatever the specific problem is?

Reply: It is an open question on the high-altitude energy source for the acceleration of inverted-V electrons and the associated quasi-static potential drop. To our knowledge, there is no competing ideas on this. In addition, as Reviewer 3 pointed out, “I am not aware of an event oriented, careful evaluation of the auroral M-I coupling energetics, as performed in the paper.”

(3) What is it about the observations presented here that allows the present authors to do better?

Reply: We found a fortuitous event when coordinated observations are available at platforms above and below the auroral acceleration region, with simultaneous auroral images in both hemispheres. The coordinates observations allow us to identify the existence of the quasi-static potential drop and the high-altitude kinetic Alfvén wave along the flux tube threading the discrete arc. In addition, the state-of-art instrumentations allow us to measure the key energy fluxes along the flux tube to obtain a quantitative energy budget.

(4) Is it really surprising that changes in the magnetosphere are caused by Alfvén waves propagating through the system?

Reply: It is somewhat surprising that Alfvén waves is the high-altitude energy source for the quasi-static potential drop, because Alfvén waves are previously thought to power another acceleration mechanism, i.e., the acceleration of the broadband electrons. The interplay between the two mechanisms (potential drop and Alfvén wave) is barely mentioned in previous literature. Now the direct observations reveal their relation from the perspective of energy budget, we believe the results would highlight the importance of interactions between the two mechanisms in future studies in the field.

(5) Why does resolving the proposed mystery matter, what are the implications for this field and beyond?

Reply: Our conclusion places critical observational restrictions on existing theories on how quasi-static potential drops form and are sustained in collisionless plasmas. If Alfvén waves power both the acceleration of inverted-V and broadband electrons, an immediate next question for future studies is what controls whether Alfvén waves power which type of acceleration. As shown on both Earth and Jupiter, inverted-V and broadband electrons are often observed in adjacent regions or even simultaneously

[Paschman et al, 2003; Mauk et al., 2017], the controlling factors are key to better understand the particle acceleration in plasmas.

We very appreciate the reviewer's high-level guidance in improving the framework of the presentation, and we believe the improved article can well present the importance of the discovery to readers. Thanks for reconsidering our revised manuscript for a publication.

Abstract: There is very little content here. See above comments. Most of the questions I list should have been answered by the end of reading a revised version of the abstract.

Reply: Thanks for the above comments, which provided a nice guidance for the improvements in our revised manuscript.

Why include Jupiter at all? Seems done for the sake of it, without a clear rationale. This physics likely happens in some form at every planet, in the Solar System and beyond, so why only show some data from Juno at Jupiter? I suggest limiting the scope of the manuscript to the Earth system, highlighting implications for other systems in the discussion.

Reply: Across the solar system, the Earth, Saturn and Jupiter are three planets often compared for obtaining a universal law in governing the auroral processes. One recent example is the controller of global auroral morphology (Zhang et al. 2024). Similarities are not always as expected, unexpected differences are crucial in revealing fundamental physical processes. We would like to introduce a recent example benefited from such comparison. Earth's magnetospheric cusp has been extensively studied for decades, and it is often believed that similar cusp shall exist at other planets such as Saturn and Jupiter, since their magnetospheres are also consequences of solar wind interaction with dipole-like magnetic field. However, the recent Juno observations identified cusp in the nightside, highlighting the importance of planetary rotation in revising magnetospheric structure (Xu et al. 2024), which was not expected in the past. To date, Jupiter is the only other planet with direct observations of the auroral acceleration region, and it was a big discovery that Jupiter's auroral particles show similar structures as the Earth (Mauk et al. 2017). Universal law. Therefore, a comparison may provide valuable implication to Jovian auroral community, broadening the impact of the results obtained from terrestrial community.

Reference

Mauk, B., et al. (2017). "Discrete and broadband electron acceleration in Jupiter's powerful aurora." *Nature* 549(7670): 66.

Zhang, B., Yao, Z., Brambles, O.J. et al. (2024). A unified framework for global auroral morphologies of different planets. *Nat Astron* **8**, 964–972.

Xu, Y., Arridge, C. S., Yao, Z. H. et al. (2024). In situ evidence of the magnetospheric cusp of Jupiter from Juno spacecraft measurements. *Nature communications*, 15(1), 6062.

Figure 1: Poor quality, needs work to improve clarity of axes labels, etc.

Reply: We have updated Figure 1 with improved quality, to show the inverted-V signatures and the relevant labels.

Reviewer #2 (Remarks to the Author):

Review of manuscript “The energy source powering aurora through a static electric potential” by S. Tian et al.

From an auroral event on 16 April 2015, the authors evaluate data simultaneously obtained with four different means at three different altitudes in a unique constellation between satellites and aurora. The data covered electric and magnetic fields measured by the two radiation Belt Storm Probes, RBSPs, near 5.8 RE, electron fluxes measured by the Defence Meteorological Satellite Program, DMSP, at 840 km height, and observations of all sky cameras on the ground. The authors focus on the energy flow into the acceleration region of the auroral particles, differentiating between electrostatic and wave acceleration. The paper is composed of a Main part containing the results and a Method part meticulously describing how the data were obtained and analysed. The central quantity to be derived was the Poynting flux and its comparison with the flux of the electrons and upward accelerated O⁺ ions. The main conclusion was that kinetic Alfvén waves, excited in the source region encountered by the RBSP spacecraft, were carrying the energy into the acceleration region between 2 and 4 RE altitude with sufficient power to cope for the observed particle fluxes. The paper is most interesting. The separation into main results and methods is very beneficial for the reader and illuminating for the expert. All parameters and the instrumentation are well described. Much emphasis is placed on the identification of the waves, their spectra, and energy fluxes. A particularly nice piece of work is the determination of the height of the acceleration region from the dispersion of O⁺.

Reply: We appreciate the reviewer’s positive general comment about the study, and we much appreciate the reviewer’s constructive suggestions for further improvements. Below are our one-to-one responses to the comments/suggestions.

I have a few reservations about two conclusions. (1) The authors suggest that the Poynting flux carried by kinetic Alfvén waves is absorbed in the auroral acceleration region (lines 117/118, 142/143, 171/172, 535/536), admitting, however, that they are unable to propose a mechanism. It is commonly thought that kinetic or inertial Alfvén waves in the magnetosphere have frequencies higher than shown in Figure 3 and are absorbed at low heights generating strongly field-aligned electrons with a diffuse energy spectrum. By contrast, the Poynting flux derived from the RBSP wave data is dominated by the low frequencies. These are most likely an expression of the plasma dynamics at the source region, as indicated by the strong changes in tilt angle of the magnetic field (Figure 2d). The main effect of the agitated plasma is bending or shearing of the magnetic field, which is transported along the field lines towards the ionosphere by regular, not kinetic Alfvén waves.

Reply: We agree with the Reviewer’s comment. The confusion arises from the terminologies. Here, we agree that the Poynting flux is carried mostly by the “regular” Alfvén waves, i.e., the MHD Alfvén waves in common textbooks. In MHD, parallel E field is strictly 0, and that both kinetic Alfvén waves and inertial Alfvén waves are “degenerate”, i.e., they both reduce to the MHD Alfvén wave. However, in strong events like the one shown in this paper, instead of $E_{\parallel} = 0$, a more proper assumption is $E_{\parallel} \ll E_{\perp}$. A finite but small E_{\parallel} could exist, even at the MHD scales at low frequencies. Consequently, the kinetic Alfvén waves and inertial Alfvén waves are not degenerate. This is illustrated by Figure 1 in Lysak 2023 that we cited below, which reprints the same results in Lysak and Lotko 1996.

Fig. 1 HYPERLINK "sps:id::fig1|locator::gr1|mediaobject::0"Dispersion relation for the kinetic Alfvén wave as a function of $k_{\perp}c/\omega_{pe}$ and v_e^2/V_A^2 . **a** Parallel phase velocity normalized by the Alfvén speed; **b** Landau damping rate normalized by wave frequency; **c** ratio of parallel electric field to the perpendicular electric field, normalized by the ratio of parallel to perpendicular wave numbers; **d** ratio of electric field to magnetic field normalized by the Alfvén speed. (after Lysak and Lotko 1996; Lysak 1998)

In this Figure 1a, we can see that kinetic Alfvén waves are above the contour line of $v_{\text{phase}}/v_A = 1$ and inertial Alfvén waves are below that line. Therefore, at MHD scales, i.e., large scale and thus small k_{\perp} , although both kinetic Alfvén waves and inertial Alfvén waves have a parallel phase velocity that is close to the Alfvén speed, according to the kinetic theory, they can still be separated by plasma beta. Here, we examined the local plasma beta around the RBSP spacecraft and showed that it is in the beta range for the

kinetic Alfvén wave ($1 > \beta > m_e/m_i$). This applies to all scales, including the “MHD scales”. For this reason, the term “kinetic Alfvén waves” in this paper refers to both the “regular” Alfvén waves (Alfvén waves at MHD scales) and the “kinetic” Alfvén waves (Alfvén waves at kinetic scales). We have clarified the description in the revised manuscript. Please see Lines 520-526.

Lysak, R. L. (2023). Kinetic Alfvén waves and auroral particle acceleration: a review. *Reviews of Modern Plasma Physics* 2023 7:1, 7(1), 1–28. <https://doi.org/10.1007/S41614-022-00111-2>

Lysak, R. L., & Lotko, W. (1996). On the kinetic dispersion relation for shear Alfvén waves. *J. Geophys. Res.*, 101(A3), 5085–5094. <https://doi.org/10.1029/95JA03712>

On the other hand, the concentration of the wave energy flow into downward and westward direction is surprising. Some comments on that may be helpful. With respect to the interaction with the AAR, I find the paragraph from lines 176 to 187 not very illuminating on this subject.

Reply: As pointed out by the Reviewer, the main wave energy is carried by the “regular” Alfvén waves, i.e, the Alfvén waves at MHD scales. This wave energy is mostly transmitted along the magnetic field lines toward the ionosphere. The westward development of the auroral arc (Figure 3a) is likely to be an indication of the westward development of the source region of the regular Alfvén waves. In other words, the spatial development of the auroral arc is a “projection” of its source region’s development in the magnetosphere. The auroral arc and its source region is connected by the regular Alfvén waves, on the order of Alfvén waves’ transit time (on the order of 10 sec for inner magnetosphere). This connection seems to be well supported by the time-lagged dipolarizations observed by the twin RBSP satellites (Figure 2d). The westward satellite (RBSP-B) saw the Alfvén waves and the associated dipolarization about 2.5 minutes later than RBSP-A (Figures 3b-3e).

For the paragraph that is originally in lines 176-187, we have added more explanations, following the Reviewer’s inspiring comments. The added texts are in Lines 191-195.

(2) The derivation of the relative velocity of spacecraft and plasma from Figure 3a is rather questionable. The narrow, elongated shape of the arc suggests that it is generated from a boundary. The variation of the brightness along this boundary likely reflects variations of the

agitation from higher altitude, as for instance by a BBF. However, I do not see how much this affects the conclusions.

Reply: The derivation of the relative velocity of spacecraft and plasma is based on previous studies showing that the dipolarization's azimuthal motion reflects the plasma's azimuthal velocity [Angelopoulos et al., 2008, Ogasawara et al., 2011]. One limitation with the RBSP dataset is that the published data products of the particle instrument (HOPE) data do not include the ion velocity. Therefore, the best estimate for the plasma velocity is from the timing of dipolarization (Figure 2d) and/or auroral's motion (Figure 3a).

Angelopoulos, V., Sibeck, D., Carlson, C. W., McFadden, J. P., Larson, D., Lin, R. P., Bonnell, J. W., Mozer, F. S., Ergun, R., Cully, C., Glassmeier, K. H., Auster, U., Roux, A., LeContel, O., Frey, S., Phan, T., Mende, S., Frey, H., Donovan, E., ... Sigwarth, J. (2008). First Results from the THEMIS Mission. *Space Science Reviews*, 141(1), 453–476. <https://doi.org/10.1007/s11214-008-9378-4>

Ogasawara, K., Kasaba, Y., Nishimura, Y., Hori, T., Takada, T., Miyashita, Y., Angelopoulos, V., Mende, S. B., & Bonnell, J. (2011). Azimuthal auroral expansion associated with fast flows in the near-Earth plasma sheet: Coordinated observations of the THEMIS all-sky imagers and multiple spacecraft. *Journal of Geophysical Research: Space Physics*, 116(A6). <https://doi.org/10.1029/2010JA016032>

Overall, a good paper, worth to study. Can be published essentially as is, after response to my comments.

Gerhard Haerendel

Reply: Thank you very much Prof. Haerendel for the guidance on further improvements. We hope the revised manuscript can be considered for a publication.

Reviewer #3 (Remarks to the Author):

The paper "The energy source powering aurora through a static electric potential" addresses the energy transport / balance in the auroral magnetosphere-ionosphere (M-I) by presenting an event study based on Van Allen / RBSP, DMSP F19, and ground THEMIS optical data. While the fundamental components of the energetics involved are understood nowadays rather well, close quantitative estimates are scarce, if at all. In this respect, the manuscript fills a significant empty spot and my overall assessment is that, essentially, the paper is suitable for publication in Nature Communications. However, I think the message can be better streamlined and the contribution made by the paper better pinpointed, as detailed below.

Reply: We much appreciate the reviewer's positive general comment and the constructive comments/suggestions on the further improvements. Below are our one-to-one response to your comments.

1. The authors emphasize the 'mystery' around auroral acceleration and particle energization, related to the relationship between quasi-static and Alfvénic acceleration (L27, 39, 45, 50, 203), as well as the 'discovery' made. I feel a bit hesitant on using those words, since the matter has been broadly addressed in the literature - as proven also by the rich selection of cited papers. The fundamental building blocks were summarized in several monographs (e.g., Auroral Plasma Physics, Springer, 2003, <https://doi.org/10.1007/978-94-007-1086-3>) and quantitative evidence for the conversion of the Alfvénic Poynting flux into particle energy flux was brought, e.g., in statistical studies based on Polar and FAST data (Keiling et al., 2002, <https://doi.org/10.1029/2001JA900140>, Chaston et al., 2002, <https://doi.org/10.1029/2002JA009272>, Chaston, 2006; in particular Figure 2, <https://doi.org/10.1029/169GM16>). These being said, I am not aware of an event oriented, careful evaluation of the auroral M-I coupling energetics, as performed in the paper - including Poynting flux above the acceleration region, precipitating electrons, auroral emissions, and outflowing ions. To my understanding, the importance of the paper resides mainly in this thorough quantitative analysis.

Reply: Exactly as pointed out by the reviewer, to our knowledge, there has not been a study on the energy conversion processes associated with the auroral acceleration region. Besides the thorough quantitative analysis, our event also shows the close correlation between the kinetic Alfvén waves at high-altitude and the quasi-static potential drop inferred from the low-altitude observation of inverted-V electrons and the ground observation of the discrete arc. Although previous statistical studies (Keiling et al., 2002; Chaston et al., 2002; Chaston 2006) suggest the important role of Alfvén waves in powering the auroral acceleration, whether the Alfvén waves power the quasi-static acceleration is still a "mystery". Here, our

event suggests that this is the case on Earth and potentially applicable on Jupiter, which is the only other planet that human being has direct measurements of inverted-V electrons. Following the reviewer's comment, we have also highlighted the importance of quantitative analysis in the revised abstract.

2. The authors emphasize as well the universal nature of auroral acceleration, which plays an important role in various environments, within and beyond the solar system. Similar to the processes responsible for auroral acceleration, this matter was addressed too in the past (e.g., AGU Monograph 197, *Auroral Phenomenology and Magnetospheric Processes: Earth and Other Planets*, <https://doi.org/10.1029/GM197>), and the importance of the parallel electric field (Alfven, 1958, <https://onlinelibrary.wiley.com/doi/abs/10.1111/j.2153-3490.1958.tb01991.x>) was emphasized also for astrophysics (e.g., Haerendel, 1994, <https://adsabs.harvard.edu/pdf/1994apjs...90..765h>). On the other hand, the evidence in the paper builds on the event of April 16, 2015, observed at the Earth by conjugate satellite and ground data. This should be made clear upfront. Unlike the Earth panel of Figure 1, which is detailed and developed in the paper (including Extended data Figure 2), there is no follow-up to the Jupiter panel of Figure 1. Merging the Earth panel of Figure 1 with Figure 2 may help the streamlining, while the Jupiter panel could be added as a small inset, for comparison (or eliminated).

Reply: The Earth and Jupiter are the only two planets on which inverted-V electrons have been observed (Figure 1). This is key evidence suggesting that the auroral acceleration through a quasi-static potential drop is a universal plasma process. Here, we take advantage of the comprehensive observational platforms on Earth but it is highly likely that the results are applicable to Jupiter's magnetosphere and other types of plasmas with quasi-static potential drops.

3. The authors refer to the 'arc' geometry of the aurora both in Figures 2-a ('Arc north'), 2-b ('Arc south'), in the caption of Figure 3-a ('... the auroral arc during its azimuthal development'), and in Figures 4-a, 4-b:

3a. The arc geometry is rather clearly visible in the DMSP F19 SSUSI data of the southern aurora, but less clear in the THEMIS ASI ground data of the northern aurora - even though, the 'skeleton' of the aurora looks, indeed, rather similar. Perhaps this skeleton can be better emphasized in Figures 2-a and 2-b, e.g., by adding some zoom-in sub-panels?

Reply: The ASI image unfortunately looks less clear because of two reasons: (1) we "merged" the data from two adjacent all-sky imagers; (2) there were thin clouds covering a small portion of one of the all-sky imager's field of view. We have tried some adjustments on the color scale but do not get a better result for the above reasons.

3b. The arc development of 4 deg/min is hard to follow in Figure 3-a. Please improve. It would be nice to add a panel similar to 3-a for the southern aurora, based on F19 SSUSI data - to pinpoint, perhaps, the similar dynamics in the two hemispheres (related to the similar skeleton under 3a?).

Reply: For Figure 3-a, we added a thicker line and an arrow to guide reader's eyes for the arc development of 4 deg/min.

Unfortunately, instead of a series of images, DMSP SSUSI only provides a snapshot of the auroral oval over about 20 min. Therefore, the arc's spatial and temporal development can only be shown in the northern hemisphere. Here the comparison in Figure 2-a and 2-b is meant to show that the arc was seen symmetrically in both hemispheres. In addition, combining Figure 1a and Figure 2b, DMSP measurements demonstrate that the inverted-V electrons were observed right over the discrete arcs, suggesting the existence of the quasi-static potential drop above the arcs.

3c. I do not understand the negative ASI counts in the color scales of Figures 2-a and 3-a (which does not seem to matter to the figures, since all the counts are bluish / positive). Perhaps the negative counts are related to outflowing energy fluxes (similar to the energy flux calibration of the SSUSI data?), but if so, this should be explained. In terms of ASI raw data, I guess the counts can only be positive.

Reply: Sorry for the confusion, the original counts of ASI are all positive. We used a symmetric dynamic range (same negative and positive ranges) and color table because they are used in our event search. This is meant to capture nonphysical data because they will show up as reddish colors. For the ASI, we used a documented background removal algorithm [Tian et al., 2023, Appendix] to better pull out the auroral forms. The algorithm may result in some small negative count values.

Tian, S., Wang, C.-P., Lyons, L. R., Bortnik, J., Weygand, J. M., Liu, J., Yadav, S., Ma, Q., Reeves, G. D., Henderson, M. G., & Wolf, R. A. (2023). Multiple Auroral Streamer Bundles in Conjugate Hemispheres and Multiple Near-Earth Injections. *Journal of Geophysical Research: Space Physics*, 128(1), e2022JA031010.
[https://doi.org/https://doi.org/10.1029/2022JA031010](https://doi.org/10.1029/2022JA031010)

4. The dynamics and geometry of the observed auroral event deserve more attention. The energy fluxes of 300+ mW/m² (Poynting) and 100+ mW/m² (electrons) are remarkably

high, compare also with Figure 2 of Chaston, 2006 (see point 1). Likewise, the energy flux of 87 mW/m^2 is well above the visibility threshold (empirically $\sim 1 \text{ mW/m}^2$). Such energy fluxes are characteristic for the most dynamic auroras, which tend to develop as 2-D forms, rather than 1-D arcs. The association of high energy fluxes, specific to very dynamic events, with arc aurora, is quite remarkable and it might be insightful to see also the dynamics behind, as captured by the THEMIS ASI and DMSP F19 SSUSI (e.g., movie links to some open access repository?), in addition to the pdf still images. Indications on the geophysical indices (Kp, Dst, AE,...) would be helpful as well, to introduce the context of the event - a rather disturbed day, during an extended moderate storm.

Reply: The several 100s mW/m^2 Poynting flux is very typical at the altitude above AAR. We have dozens of other similar events from RBSP and THEMIS. We typically see such high Poynting flux when the spacecraft were in conjunction with bright discrete arcs. We have planned future works to study these very dynamic events. The arcs in such events are still in general 1D. But if the observational resolution allows, there will probably be smaller scale 2D structures, as suggested by previous studies (e.g. Knudsen et al., 2001).

We didn't include information on Dst, AE, and Kp because these geomagnetic indices are specifically used by the space physics community and thus might be too detailed for a more general audience. However, the reviewer is right that the event on April 16, 2015 was during the main phase of a moderate geomagnetic storm (Dst minimum of -88 nT). At the time of the arc, the Dst was around -80 nT . To accommodate the reviewer's suggestion, we added the following sentence in Line 68-69: "This event occurred during an extended moderate geomagnetic storm."

Knudsen, D. J., E. Donovan, L. Cogger, B. Jackel, and W. Shaw: 2001, 'Width and structure of mesoscale optical arcs' . Geophys. Res. Lett. 28, 705.

5. Others

- L87 and Figure 2-c: The tilt angle, θ , is defined as the absolute value of $\sin^{-1}(B_z/B)$ (missing vertical bar, for absolute value, at the left side of this expression?). Based on Figure 2-c, where the tilt angle is rather small, between 20 and 40 deg, and on the satellites' location near the equatorial plane, the 'tilt angle' appears to be the angle made by the magnetic field with the z axis (equal to zero at the equator, in an ideal dipole configuration). If so, this angle should be $\cos^{-1}(B_z/B)$.

Reply: This is a typo. The tilt angle is $\sin^{-1}(B_z/|B|)$. There is an extra "l" that should be removed.

- L95-96: The tilt angle is now called elevation angle. Please keep to the same name, be it tilt or elevation. Apart from that, it is noted the increase at RBSP A and B, separated by a couple of minutes, but there is no comment on the simultaneous change, at ~08:07:50, from a slow decrease in both RBSP A and B, to a rapid increase in RBSP A and a rapid decrease in RBSP B (before the increase at 08:10). Any hint on what could be the reason behind? (perhaps in terms of BBF configuration, L192-196?) The increase of the tilt / elevation angle is associated with the release of magnetic energy. Is this an indication to (BBF related?) dipolarization? If yes, the tilt / elevation angle is rather made with the x-y (equatorial) plane (contrary to my inference above). Please clarify.

Reply: We now uniformly call it the elevation angle for consistency. The elevation angle is the angle in terms of the x-y (equatorial plane).

In terms of what happened around 08:07:50 to 08:10 UT, from the shown data, we infer that there was a global scale stretching, reducing the elevation angle of the Earth's magnetic field lines at all MLTs. Then there was a dipolarization, which may be caused by a BBF but we cannot tell since the ion velocity data is not available for RBSP. Nevertheless, this dipolarization first arrived at RBSP-A around 08:07:50 UT, causing the local magnetic field to dipolarize (the sharp increase of the elevation angle). In the meantime, the global-scale stretching continues at regions outside the dipolarization, e.g. at RBSP-B. At 08:10 UT, the dipolarization propagate westward and arrived at RBSP-B, causing the local magnetic field to dipolarize.

- L100: RBSP A and B sample the inner magnetosphere (as noted at L189), rather than the 'distant magnetosphere'. As a compromise, one could leave just 'magnetosphere'.

Reply: Corrected.

- L112, 114, 155, 156, 530-536 and Figure 4-b: The (maximum) energy fluxes for electrons and oxygen ions are indicated in the text as 162 and 44 mW/m², respectively, whereas Figure 4-b shows ~165 and ~50 mW/m². I understand the approximation, but it might be good to have the same numbers everywhere (also since the approximation is not applied to the 87 mW/m² of the auroral emissions). At L155 the energy flux of the escaping O⁺ is indicated wrongly as 87 mW/m². Should be changed to 44 and, accordingly, the efficiency at L156 should be $(162 + 44) / 350 = 59\%$, instead of 72 %. Same efficiency of 59% at L536, instead of 57%.

Reply: We thank the Reviewer for the detailed corrections on typos, grammar, notations, and etc. Now have consistent numbers for the energy fluxes throughout the texts and figures.

- L119, 120: *In line with point 2, please replace 'planet' with 'Earth'. Certainly, there is ion outflow also at planets, but this very event is observed at Earth (and all the numbers characterize this event).*

Reply: Corrected.

- L144-146: *Please explain this consistency. See also point 3b above.*

Reply: We deleted this sentence here and focus on discuss the consistency at the changes related to point 3b.

- L148-149: *Not sure I understand this sentence. Do you mean that the Poynting flux is mainly contributed by frequencies under 100 mHz, which correspond to shear Alfvén wave regime, according to panels a) and c) of Extended data Figure 1 (E/B ratio around v_A)?*

Reply: Sorry for the confusion. Yes, from the E/B ratio in the Extended data Figure 1, it happens to be that the spin fit resolution covers the “shear Alfvén wave”. If we integrate the power of the Poynting flux over frequency, then it is the case that the shear Alfvén waves made the majority of the contribution.

However, here we are discussing the peak value of the Poynting flux, which depends on the frequency range included. As shown in Figure 3, the peak values (magnitude) of the spin fit resolution electric field at both spacecraft were about 60 mW/m² (Figures 3c and 3e). However, the more proper peak value is 350 mW/m², using the higher resolution electric field, which unfortunately only available at RBSP-B (Figure 3e).

To be more specific, we rewrite the relevant sentences as follows: “The Earthward Poynting flux is about 60 mW/m² as calculated from the spin fit electric field (up to ~100 mHz) at both satellites (red lines in Panels c) and e) in Figure 3). However, comparing the frequency spectrogram of the magnetic field at RBSP-A (Figure 3b) to that at RBSP-B (Figure 3d), we note that the wave power at the smaller kinetic scales does exist at both RBSP-A and -B (>100 mHz). The availability of higher cadence electric fields at RBSP-B allows us to include this power, resulting in a better estimate of the actual Earthward Poynting flux (Figure 3e orange line).”

- L196-197: *The relationship between most intense Alfvén waves and density gradients is not addressed in the paper.*

Reply: We observed that the most intense Alfvén waves are around the density gradients in this event as well. However, since this is previously observed and simulated [Tian et al., 2021 Figure 2; Lysak and Song 2011 Figure 1], we did not show the observations in the main texts or Methods.

Now we rephrase this sentence as follows: “As these Alfvén waves tend to be more intense around density gradients [Tian et al., 2021], it seems to favor the generation through mode-conversion [Lysak and Song 2011].”

Tian, S., Colpitts, C. A., Wygant, J. R., Cattell, C. A., Ferradas, C. P., Igl, A. B., Larsen, B. A., Reeves, G. D., & Donovan, E. F. (2021). Evidence of Alfvénic Poynting Flux as the Primary Driver of Auroral Motion During a Geomagnetic Substorm. *Journal of Geophysical Research: Space Physics*, 126(5), e2020JA029019.
<https://doi.org/https://doi.org/10.1029/2020JA029019>

Lysak, R. L., & Song, Y. (2011). Development of parallel electric fields at the plasma sheet boundary layer. *Journal of Geophysical Research: Space Physics*, 116(A1), n/a--n/a.
<https://doi.org/10.1029/2010JA016424>

- L483-484: *Why this association, between the flow speed and the azimuthal development of the arc?*

Reply: This is based on previous studies [Angelopoulos et al., 2008, Ogasawara et al., 2011], showing that the azimuthal propagation of a dipolarization scales to the plasma flow. We have added these references in Line 500.

Angelopoulos, V., Sibeck, D., Carlson, C. W., McFadden, J. P., Larson, D., Lin, R. P., Bonnell, J. W., Mozer, F. S., Ergun, R., Cully, C., Glassmeier, K. H., Auster, U., Roux, A., LeContel, O., Frey, S., Phan, T., Mende, S., Frey, H., Donovan, E., ... Sigwarth, J. (2008). First Results from the THEMIS Mission. *Space Science Reviews*, 141(1), 453–476.
<https://doi.org/10.1007/s11214-008-9378-4>

Ogasawara, K., Kasaba, Y., Nishimura, Y., Hori, T., Takada, T., Miyashita, Y., Angelopoulos, V., Mende, S. B., & Bonnell, J. (2011). Azimuthal auroral expansion associated with fast flows in the near-Earth plasma sheet: Coordinated observations of the THEMIS all-sky imagers and

multiple spacecraft. *Journal of Geophysical Research: Space Physics*, 116(A6).
<https://doi.org/10.1029/2010JA016032>

6. Typos and alike

L31: *is launched from => resides in (?)*;

Reply: we have rewritten it as: *Energy carried by kinetic Alfvén waves travels from the magnetosphere to the auroral acceleration region and forms electric potential drop that further acceleration particles to produce aurorae.*

L111: *Figure 1b => 1a*;

L144: *transverse => traversal (?)*;

L145: *crossingS*;

L146: *shown IN*;

L168: *powerS*;

L176: *on Earth => above the Earth*;

L211: *crystal => crustal*;

L324: *Mhd => MHD* (use {MHD} with bibtex, likewise for all capital letters / acronyms);

L478: *Full stop after ω_{ci} (?)*. Then: *The velocity of kinetic Alfvén waves can be approximated... (?)*;

L485: *we infer this is => we infer that (?)*

L492: *Panel a-1 => Panel a*;

L499: *suggestING*;

L523, L524: *Panel b-2 => Panel b of Extended data Figure 2*;

L525: *correlated => correlates (?)*;

L543: *O+ => O+ velocity*;

L570: *the timing ... IS consistent.*

Reply: All corrected, thanks.

Response to the Referees' comments

We thank the referees for their constructive comments and suggestions for our first revision. Again, we appreciate their reviews to help us to revise and improve the manuscript. Below are our point-by-point responses to the referees' remarks. The original referees' comments are quoted in *italics* and our responses are in blue for convenience. All requested revisions have been made according to the referees' suggestions.

Reviewer #1 (Remarks to the Author):

The authors have addressed all my questions. Although the manuscript presents solid, original findings, I am of the opinion that this is an incremental advance, without substantial implications for the field. Perhaps a different journal would be better fit, but I am not well qualified to judge.

We are pleased that Reviewer 1's questions have all been addressed by our previous revision. We thank Reviewer 1's effort in evaluating the manuscript and the high-level guidance in shaping the manuscript's organization.

Consistent with the evaluations from the other reviewers, we list the importance of our energy budget analysis.

Firstly, our auroral conjunction event, when multi-platform observations at all key altitudes become in conjunction along the same auroral magnetic flux tube, is unprecedented and rare on Earth and impossible in the foreseeable future on other planets or astronomical objects. In analogy, a recent conjunction along the solar wind magnetic flux tube is reported in Science (Rivera et al., 2024). The rareness of such conjunctions naturally makes them significant contributions.

Secondly, the topic of our conjunction event, i.e., the auroral acceleration, is among the several most important long-lasting open questions in space plasma physics. Like the Science paper (Rivera et al., 2024), which studies the energy budget along the solar wind magnetic flux tube and sheds light on how solar wind is heated, our analysis on the auroral energy budget reveals key information on what powers the auroral acceleration and how, which has been explained in our previous reply.

Thirdly, the auroral acceleration as a universal feature of space plasma, has been accepting increasing attention from planetary scientists and astrophysicists, given

the verification of the inverted-V electrons in Jupiter and the similar optical measurements on other planets and even exoplanets. Our understanding of auroral physics has been significantly **broadened** based on comparison/parameterized studies on the auroral forms on different planets. As Earth magnetospheric scientists, we have the unique advantage of the availability of multi-platforms that are only available on Earth. For this reason, the Earth is the only place we can significantly **deepen** our understanding of auroral physics. Our study on the auroral energy budget is an example of the unique contribution that can be made from the Earth magnetosphere community to the broad audience including the planetary scientists, astrophysicists, plasma physicists, etc.

Fourthly, our study will have a broad impact on future studies and spacecraft missions. For example, the energy budget places key observational information in testing existing and future theories and simulations on Earth, Jupiter, and other astronomical objects. Our energy budget highlights the key role of Alfvén waves and will affect future studies to explore how Alfvén waves evolve into the quasi-static potential drop. We believe this work will inspire a substantial advance in our understanding of how electron acceleration occurs in general in the universe. In addition, to resolve the rareness of good conjunction events like the one we showed here, a combination of multi-spacecraft at key altitudes and ground platforms is the only solution. Such comprehensive missions will not only be able to study auroral physics, but also other key processes in the Earth's magnetosphere, ionosphere, and thermosphere.

Again, we appreciate the reviewer's comments that motivated us to improve the presentation of the manuscript.

References

Rivera, Y. J., Badman, S. T., Stevens, M. L., Verniero, J. L., Stawarz, J. E., Shi, C., Raines, J. M., Paulson, K. W., Owen, C. J., Niembro, T., Louarn, P., Livi, S. A., Lepri, S. T., Kasper, J. C., Horbury, T. S., Halekas, J. S., Dewey, R. M., de Marco, R., & Bale, S. D. (2024). In situ observations of large-amplitude Alfvén waves heating and accelerating the solar wind. *Science (New York, N.Y.)*, 385(6712), 962–966. https://doi.org/10.1126/SCIENCE.ADK6953/SUPPL_FILE/SCIENCE.ADK6953_SM.PDF

Reviewer #2 (Remarks to the Author):

What are the noteworthy results? A concept of powering the auroral acceleration by kinetic Alfvén waves.

Will the work be of significance to the field and related fields? How does it compare to the established literature? If the work is not original, please provide relevant references. The work is original and will be significant through the outstanding data set.

Does the work support the conclusions and claims, or is additional evidence needed? Yes, it does.

Are there any flaws in the data analysis, interpretation and conclusions? Do these prohibit publication or require revision? There are problematic interpretations. No revision required.

Is the methodology sound? Does the work meet the expected standards in your field? Yes.

Is there enough detail provided in the methods for the work to be reproduced? Yes.

We thank Reviewer 2 for the evaluation on the overall importance of this study.

2nd 1 Review of manuscript “The energy source powering aurora through a static electric potential” by S. Tian et al.

I maintain my previous judgment that this is an interesting paper worth publishing. However, I now have some hesitation to add the words “as is”. The point is that the authors did not react properly to my comment: “These are most likely an expression of the plasma dynamics at the source region, as indicated by the strong changes in tilt angle of the magnetic field (Figure 2d).” In their reply, the authors say after Figure 1: “Here, we examined the local plasma beta around the RBSP spacecraft and showed that it is in the beta range for the kinetic Alfvén wave ($1 > \beta > m_e/m_i$ “. However, the fact that the deviation of tilt angle from the background magnetic field is of the same magnitude as that field value suggests that there is a background plasma present with $\beta \sim 1$ and the condition for kinetic Alfvén waves: $\beta \ll 1$ is not fulfilled.

There are two conflicts, one about the proper value of beta and the other about the proper designation of the waves constituting the Poynting flux with peak at 10 mHz

as kinetic Alfvén waves (Lines 516-518).

Conflict 1: The authors say in lines 520-521: "...the term kinetic Alfvén wave refers to the shear-mode Alfvén waves at $m_e/m_i \ll B \ll 1$ ".

Conflict 2: In lines 516-518: "Panels b and d show that the power of the Poynting flux associated with the kinetic Alfvén waves peaks around the 10 mHz frequency range".

We thank Reviewer 2 for raising the concerns related to the plasma beta and the role of the Alfvén waves around 10 mHz. We think the two conflicts are caused by the terminology of "kinetic Alfvén waves" we used in the last version, which we now have been fixed. Please check the details below.

The plasma beta for this event is $\ll 1$ because of the strong background magnetic field in the inner magnetosphere. As shown in the plot below, although significant changes of the tilt/elevation angle (Panel a), the background magnetic field is large (magnitude about 180 nT, Panel b), leading to $\beta < 0.1$ throughout the event (Panel c). Beta is calculated as the ratio of the thermal pressure (electron, H⁺, and O⁺ ions) over the magnetic pressure shown in Panel d). This is why we refer to the Alfvén waves for all frequencies using the terminology "kinetic Alfvén waves" (KAWs).

We think that the Alfvén waves below and above 10 mHz play different roles. The E_{\parallel} is mostly carried by those above 10 mHz (when E/B ratio starts to be much larger than v_A). Because the beta remains below 0.1 throughout the event, we tried to call both KAWs for simplicity to the more general audience of Nature journals. We attempt this simplification because the main focus is the energy budget instead of discussing the different roles the different components of the Alfvén waves play.

Inspired by the comments from Reviewer 2, we now adopted two major changes: (1) Refer to the Alfvén waves below and above 10 mHz as the "regular" Alfvén waves and KAWs, respectively. Following this change, conflict 1 is resolved because the "regular" Alfvén waves are not insensitive to plasma beta. Conflict 2 is also resolved because the power peaks around 10 mHz and corresponds to the "regular" Alfvén waves; (2) replace KAWs in key places by the more general term "Alfvén waves", when we need to refer to both wave components below and above 10 mHz.

We also note the significant change of the tilt angle of the magnetic field might arise from the fact that the tilt angle is small (15-20 deg) right before the dipolarization (around 08:10 UT). This is a natural result of a highly stretched background magnetic

field. From this small tilt angle, an increase to 30-40 deg is significant in percentage (100%).

I believe that the authors insist on the designation: "kinetic Alfvén waves", because they are thought to carry a finite $E_{||}$, and that is needed, as "Energy carried by kinetic Alfvén waves travels from the magnetosphere to the auroral acceleration region and forms electric potential drop that further acceleration particles to produce aurorae."(Lines 30-32)

To be more specific, we have changed kinetic Alfvén waves to Alfvén waves, to refer to both the Alfvén waves at MHD and kinetic scales, for the reasons explained above.

In my first review, I did not want to enter into a debate on the overall concept, which starts from an energy source in form of kinetic Alfvén waves and lets them be absorbed at lower altitudes to form an electrostatic potential drop. Now I see a need to do so.

I further question the validity of the formulation in lines 191-194:” According to our observation, the main wave energy is carried by the shearing of the magnetic fields, i.e., the “regular” Alfvén waves, transported along the field lines towards the ionosphere. The energy conversion around the AAR is likely “dispersive”, i.e., through Alfvén waves at kinetic scales.”

Alfvén waves do not shear the magnetic field. The shear is in the magnetic field and caused by the field-aligned current. The Alfvén waves carry the energy gained by unshearing the magnetic field.

We have deleted the part “shearing of the magnetic fields”, which is added to refer to the Alfvén waves below 10 mHz. This part reads as follows: “the main wave energy, carried by the “regular” Alfvén waves at magnetohydrodynamic (MHD) scales, is transported along the field lines towards the ionosphere”.

This is in brief my understanding of the energization of the AAR:

Origin is a concentrated field-aligned current. It is connected with shearing the magnetic field and thus storing energy in the field. Next is the appearance of an electric resistance to the field-aligned current at heights above about 1500 km. It may be connected with the mirror effect according to (Knight 1973). The resistance causes the presence of a parallel potential drop, which decouples the magnetic field from the ionosphere. Decoupling means that the “field lines” (i.e. plasma and field) move in a way as to remove or lower the magnetic shear. This perturbation of the field propagates upward (and downward, but less important) with Alfvén speed constituting a front behind which the sheared field is unsheared. The released magnetic energy is transported by Alfvén waves towards the potential drop thus supplying it with the energy to be consumed by energization of electrons and ions. I think that I owe the authors this summary to let them understand my above formulated criticism. However, I do not want them to modify their concept, just to take note of an alternate scenario and reply to my comments along lines 3 to 19.

We thank Reviewer 2 for the explanation of this physical picture. We think this is a possible scenario on how Alfvén waves could be generated and how auroral field lines operate. We will investigate this topic in future studies.

As the main focus of this study is the energy budget along the magnetic field lines threading the quasi-static potential drop, we are open to Reviewer 2’s discussion on the “overall concept”. We think Reviewer 2’s physical picture is indeed plausible, because this is consistent with the observation that the tilt angle of the magnetic field increases (i.e. the dipolarization), which is the classical signature of releasing of previously stored magnetic energy. The magnetic energy may be released in the

form of unshearing the magnetic fields. We think this discussion is primarily about how Alfvén waves are generated along the auroral field lines (around AAR by releasing the shear, adding shear around the equator, etc). However, all the possible generation mechanisms are consistent with our energy budget analysis that highlights the Poynting flux associated with Alfvén waves (a combination of the “regular” Alfvén waves at MHD scales and dispersive Alfvén waves at kinetic scales) is the energy source that powers the auroral acceleration through the quasi-static potential drop. The detail mechanisms on how these Alfvén waves are generated and how they power the potential drops will be the focus of future studies.

Reviewer #3 (Remarks to the Author):

Thank you for the revision. While some issues have been fixed, a few require further attention:

1. The Response details the reasons for including the JUICE panel in Figure 1, in particular since Jupiter is the only other object apart from the Earth where inverted-V electrons were observed (I thought the same held for Saturn?) However, those reasons were not included in the text. Please clarify that also in the text, such that the readers can grasp easily the message of the JUICE panel, despite the fact that further analysis concentrates just on the event at Earth.

We thank Reviewer 3 for the suggestion. We have added the following in the main text to emphasize this point. “Jupiter is so far the only other astronomical object where inverted-V electrons are directly observed apart from the Earth.”

2. The authors indicate in the Response that “The several 100s mW/m² Poynting flux is very typical at the altitude above AAR. We have dozens of other similar events from RBSP and THEMIS.” This comes somewhat at odds with former studies, like Keiling et al. (2002), in particular their Figure 5. The number of intense Poynting flux events there, well above 100 mW/m², seems rather limited, 4 out of 40, i. e., some 10%. Could this be an effect of the different orbit? Of the different data processing? Of your better statistics? Some other reason? Please comment.

We thank Reviewer 2 for the careful thoughts on the odds of large events (say for events have Poynting flux > 100 mW/m² mapped to 100 km). Although not the focus of our event study, this is one immediate question that our study would inspire. Indeed, the dozens of events we mentioned are a part of our statistical study

that is under preparation. At this stage, we do not have a solid estimate of how often such large events would occur, to be compared to the 10% from Keiling et al., 2002 results. We have systematically surveyed the entire RBSP mission (about 7 years) and a couple of months of THEMIS mission. This seems to be a larger dataset than the Polar survey by Keiling et. al., 2002. A larger data set might be why we have more events.

In addition, as suggested by Reviewer 2, we also think spacecraft orbit may be a factor. Keiling et al., 2002 used the Polar spacecraft. The benefit of the Polar dataset is that its polar orbit can ensure a pass through the plasma sheet for every orbit in the nightside MLT. The drawback is that it only provides a “snapshot” type observation of the flux tube that is in conjunction with a discrete arc. On the other hand, the drawback of satellites in equatorial orbits like RBSP and THEMIS is that they are not in conjunction with a discrete arc in a large fraction of their orbits. However, if they happen to be in the plasma sheet, they remain within the plasma sheet for hours because the satellite dwells around its apogee for hours. The benefit is that they can observe the local temporal changes that can be compared to the temporal changes of the ASI aurora to identify a conjunction over the discrete arc. Another benefit is that they can record multiple events per orbit as the flux tube of the discrete arc is stretched and dipolarized.

A third factor is that we often have multiple RBSP (-A and -B like the event we show) and THEMIS (-A, -D, and -E). As the arc moves around, we tend to see large Poynting fluxes for all satellite that the arc sweeps over. During this evolution, because multiple satellites are available, they can better record the largest Poynting flux for the evolving arc.

The data processing is unlikely to be a major factor, because as we showed below, using the 180 sec limit for the event shown in the current manuscript does not significantly change the magnitude of the total Poynting flux. Here, Panel a) shows the Poynting flux in Figure 3e. This is an integration of the wavelet results of the frequency spectrogram in Panel d) from 0.55 mHz to 4 Hz (scale to 1800 sec and 1/4 sec). Panel b) shows the same Poynting flux but the only difference is that the integration is from 5.5 mHz to 4 Hz (scale to 180 sec and 1/4 sec). Panel c) shows the difference between Panels a) and b). As we can see from the frequency spectrogram in Panel d), where the color indicates the Poynting flux waveform (green for positive and brown for negative values), the significant Poynting flux is in the frequency range from 4 mHz to 2 Hz. The wavelet analysis validates the correctness of using 180 sec as the lower limit by previous Polar studies and

simulations (e.g. Keiling et al., 2002; 2003 science; Zhang et al., 2012). Because most of the significant Poynting flux has been captured by this limit. In addition, the wavelet analysis also shows that using a smaller lower limit is valid as well. Here, as an attempt to best estimate the total Poynting flux, we have adopted 1800 sec as the lower limit. This limit is chosen to enclose the significant Poynting flux around the corresponding frequency of 0.55 mHz.

References

- Keiling, A., Wygant, J. R., Cattell, C. A., Peria, W., Parks, G., Temerin, M., Mozer, F. S., Russell, C. T., & Kletzing, C. A. (2002). Correlation of Alfvén wave Poynting flux in the plasma sheet at 4–7 RE with ionospheric electron energy flux. *J. Geophys. Res.*, *107*(A7), 1132. <https://doi.org/10.1029/2001JA900140>
- Keiling, A., Wygant, J. R., Cattell, C. A., Mozer, F. S., & Russell, C. T. (2003). The Global Morphology of Wave Poynting Flux: Powering the Aurora. *Science*, *299*(5605), 383–386. <https://doi.org/10.1126/science.1080073>
- Zhang, B., Lotko, W., Brambles, O., Damiano, P., Wiltberger, M., & Lyon, J. (2012). Magnetotail origins of auroral Alfvénic power. *Journal of Geophysical Research: Space Physics*, *117*(A9). <https://doi.org/10.1029/2012JA017680>

3. Given the major importance of the quantitative argument for the paper, please include the comment above also in the text. Along the same line, regarding the data processing, your peak Poynting flux of 350 mW/m^2 relies on a frequency range whose lower limit is 0.6 MHz (L138, 143). This is different from Keiling et al. (2002), who subtract a background averaged over 3 min, roughly equivalent to a lower limit of 5.6 MHz , i.e., one order of magnitude larger.

As shown in the plot in responding Point 2, the magnitude of the Poynting flux is not significantly affected by the lower limit of the frequency range. As shown in Panels a) and b), the peak Poynting fluxes remain to be about 350 mW/m^2 , by adopting the 0.6 MHz (to be more specific 0.55 MHz) and 5.6 MHz (corresponds to 180 sec). The maximum Poynting flux in Panel a) is 2% larger than that in Panel b).

The magnitude of the Poynting flux is more sensitive to the choice of the upper limit. For example, in Panel d), we also marked the frequency corresponding to the 11 sec spin period (about 10 mHz). We can see there are significant Poynting flux from 10 mHz to 2 Hz. This is why we need to use the survey resolution electric field to obtain the full Poynting flux.

As suggested by Reviewer 3, we now have added the comments on the integration limits of the Poynting flux, in the end of the Method “The calculation of Poynting flux”.

Note that the wavelet analysis method and the choice of integration limits have been documented in detail in Tian et al., 2021; 2022; 2024. For this reason, we did not show the frequency spectrogram of the Poynting flux in this manuscript to avoid duplication.

References

Tian, S., Colpitts, C. A., Wygant, J. R., Cattell, C. A., Ferradas, C. P., Igl, A. B., Larsen, B. A., Reeves, G. D., & Donovan, E. F. (2021). Evidence of Alfvénic Poynting Flux as the Primary Driver of Auroral Motion During a Geomagnetic Substorm. *Journal of Geophysical Research: Space Physics*, 126(5), e2020JA029019. <https://doi.org/https://doi.org/10.1029/2020JA029019>

Tian, S., Lyons, L. R., Nishimura, Y., Wygant, J. R., Lysak, R. L., Ferradas, C. P., An, X., Igl, A. B., Reeves, G. D., Larsen, B. A., & Ma, D. (2022). Auroral Beads in Conjunction With Kinetic Alfvén Waves in the Equatorial Inner-Magnetosphere. *Geophysical Research Letters*, 49(9), e2022GL098457. <https://doi.org/https://doi.org/10.1029/2022GL098457>

Tian, S., Wygant, J. R., Cattell, C. A., Dombeck, J. P., & Bortnik, J. (2024). Observations of Significant Ion Energy Outflows Associated With Cusp Ion Outflows and the Role of Poynting Flux as an Energy Source. *Journal of Geophysical Research: Space Physics*, 129(10), e2024JA032644. <https://doi.org/10.1029/2024JA032644>

3a) It would be nice to provide a brief motivation for your choice of 0.6 mHz (which is consistent with the 25 min duration of the time interval in Figure 3 - but why this duration?)

The choice of this lower limit is from the wavelet analysis results in Panel d of the plot in replying Point 2. This lower limit is chosen to be below 4 mHz, where no

significant Poynting flux is observed anymore, as an attempt to include the Poynting flux as much as possible. It happens to be comparable to the during of the time interval in Figure 3. As we noted, the choice of lower limit has been added.

3b) The lower limit of the frequency range for the wavelet spectrograms in Figure 3 and for the spectra in Extended data Figure 1 is (a bit less than) 2 mHz, corresponding to some 10 min. Any particular reason for not showing the very low frequencies, from 0.6-2 mHz (corresponding to 10-25 min)?

We have updated the Extended data Figure 1 to include the data below 2 mHz. There is no particular reason for not showing that data.

3c) I presume that Extended data Figure 1 shows cuts through spectrograms (not shown) of E/B and of S_{\parallel} , similar to the spectrograms of B_{West} in Figure 3. Please indicate the time instant(s) of these cuts (perhaps the time(s) of the maximum Poynting flux?)

The Panels in the Extended data Figure 1 are the average of each quantity (E/B, S_{\parallel}) within the two minutes around the maximum Poynting flux. For each RBSP satellite, the two minutes are marked by the vertical dotted lines in Figures 3c and 3e, respectively. Note that we've updated Figure 3 because the vertical dotted lines in the previous version is 1.5 minutes which is outdated.

We have added the following in the figure caption of Figure 3 "For each satellite, the vertical dotted lines in Panels c) and e) mark the 2 minutes right after the dipolarization time in Panel a). Detailed spectral information within the 2 minutes for the Poynting flux and E/B ratio are presented in the Extended Figure 1."

We have added the following in the figure caption of Extended Figure 1 and in the Method "The identification of kinetic Alfvén waves (KAWs)". "The quantities are averaged over the two minutes interval marked in Figures 3c and 3e, coinciding with the largest Poynting flux."

3d) Much of the intense Poynting flux is contributed by the lower frequencies (as it is with many other signals - most of the power is related to / needed for the larger scales). The one order of magnitude difference in the lower frequency limit is presumably essential for the large number of events with several 100s mW/m^2 Poynting flux, mentioned in the Response.

As we mentioned earlier in the plot responding to Point 2, the magnitude of the Poynting flux is not very sensitive to the choice of the lower limit, as long as the lower limit is below the main Poynting flux power (c.f. the frequency spectrogram of the Poynting flux in Panel d in the plot and also similar plots in Tian et al., 2021; 2022).

*4. I am aware that deriving the Poynting flux from experimental data is difficult and I understand that coming up with a precise recipe is not the main target of this paper. On the other hand, the quantitative coherence of the energy budget, as emphasized in Figure 4, would be difficult to achieve, without, at least, *some* accuracy of the recipe. An immediate example is the difference between the maximum Poynting flux based on spin fit electric field, 60 mW/m^2 , as compared to that based on higher frequency electric field, 350 mW/m^2 (a larger value which, incidentally, relies also on the frequency range $<100 \text{ mHz}$, as shown by Extended data Figure 1). Obviously, the spin fit Poynting flux cannot account for the observed particle energy fluxes. The good agreement between the (maximum) higher frequency Poynting flux and the other energy fluxes, inferred from particle and optical data, provides also a self-consistent validation for the Poynting flux recipe (with due regard to the error margin). A comment on this matter, in the text, would be welcomed.*

We agree that estimating the Poynting flux is the most complicated as compared to the other energy fluxes. We think the uncertainty of the total Poynting flux is on the order of 2%, which is the difference between integrating with the lower limits of 0.55 mHz and 5.5 mHz. The upper limit is clearly needed to be well above the spin frequency. The “procedure” to best estimate the Poynting flux is documented in previous papers. We have also added a brief description as noted in responding to Point 3, in the end of the Method “The calculation of Poynting flux”.

5. Minor

L32: acceleration -> accelerates.

Corrected.

L113: The detailed analysis of energy budget is indeed provided in Methods, but the energy budget seems to fit better with Figure 4 than with Extended data Figure 2.

The full energy budget in Figure 4 is a summary of the separate energy budgets for the auroral image and inverted-V electron in Extended data Figure 2, the Poynting flux in Figures 3c and 3e and Extended data Figure 1, and the ion outflows in Extended data Figure 3. Following Reviewer 2's suggestion, we added the following to refer readers to check out the detailed analysis in these Figures: "The detailed analysis of the energy budget is shown in Extended data Figure 2 (for the discrete arc and inverted-V electrons), Extended data Figure 3 (for the O+ outflows), and Figure 3 (for the Poynting flux)."

Response to the Referees' comments

We thank the reviewer for the insightful remarks and suggestions. Below are our point-by-point responses to the remarks. The original referees' comments are quoted in *italics* and our responses are in blue for convenience. Although mentioned by the reviewer that "*The paper can be published as is.*", we have considered the reviewer's suggestion and modified Lines 207-210.

Reviewer #2

(Remarks to the Author)

The key result of the study is: "Energy carried by Alfvén waves travels from the magnetosphere to the auroral acceleration region, forming an electric potential drop that accelerates particles to produce aurorae." However, the second acceleration mechanism, directly through kinetic Alfvén waves, has not received direct attention, although much of the electron data from the DMSP might well be the result of it. Instead, the Alfvén waves are just seen as e.m. energy carriers into the AAR. Quite appropriately, the question how the wave energy is converted into the electric potential structure is left open, the main issue being the comparisons of wave energy and the energy of the directly observed electrons.

The new element is the relation of the wave energy to a dipolarization event. The rebuttal to Reviewer #2 contains new ideas about that matter. However, they have not found direct entry into the manuscript. The sentence in Lines 207-210 : "The energy conversion around the AAR is likely "dispersive", i.e., through Alfvén waves at kinetic scales. The exact mechanism on how Alfvén waves are dissipated through the potential drop is the remaining question for future investigations." is perhaps more confusing than illuminating. The first sentence compares a process with plasma entities, whereas the second sentence suggests that this process might consist of the dissipation of the waves. Fortunately, it is left open.

Here, the key problem of the paper becomes evident. First, the waves are launched from high altitudes, possibly caused by the impact of a flow burst on the magnetosphere. Secondly, the waves are dissipated or converted in an electrostatic potential structure by an unknown process. Why not starting with the generation of an AAR by an excessive field-aligned current launched by the impact of a flow burst and the AAR triggering the inflow of energy stored in the magnetic field by the same process. Then it is irrelevant whether the Poynting flux consists of inertial MHD Alfvén waves. Much of this has already appeared in my third review.

This remark does not need a response of the authors. However, it may initiate a different formulation of the sentences in Lines 207-210.

The paper can be published as is.

We thank the Reviewer for the insightful remarks and the recommendation that the paper can be published as is. Following the summary of the reviewer that “Secondly, the waves are dissipated or converted in an electrostatic potential structure by an unknown process”, we now removed the first sentence “The energy conversion around the AAR is likely “dispersive”, i.e., through Alfvén waves at kinetic scales.”. The second sentence is kept, “The exact mechanism on how Alfvén waves are dissipated through the potential drop is the remaining question for future investigations.”. The “future investigations” certainly include the suggested scenario involving the excessive field-aligned current due to a flow burst triggering the Alfvén waves.